# Large-scale network integration in the human brain tracks temporal fluctuations in memory encoding performance

**Ruedeerat Keerativittayayut[1], Ryuta Aoki[2†]\*, Mitra Taghizadeh Sarabi[1], Koji Jimura[2,3], Kiyoshi Nakahara[1,2]\***

[1]School of Information, Kochi University of Technology, Kochi, Japan; [2]Research Center for Brain Communication, Kochi University of Technology, Kochi, Japan; [3]Department of Biosciences and Informatics, Keio University, Yokohama, Japan

**\*For correspondence:**
qqqqaokiq@yahoo.co.jp (RA);
nakahara.kiyoshi@kochi-tech.ac.jp
(KN)

**Present address:** [†]Research
Institute for Future Design, Kochi
University of Technology, Kochi,
Japan

**Competing interests:** The
authors declare that no
competing interests exist.

**Reviewing editor:** Roberto
Cabeza, Duke University, United
States

**Abstract** Although activation/deactivation of specific brain regions has been shown to be predictive of successful memory encoding, the relationship between time-varying large-scale brain networks and fluctuations of memory encoding performance remains unclear. Here, we investigated time-varying functional connectivity patterns across the human brain in periods of 30–40 s, which have recently been implicated in various cognitive functions. During functional magnetic resonance imaging, participants performed a memory encoding task, and their performance was assessed with a subsequent surprise memory test. A graph analysis of functional connectivity patterns revealed that increased integration of the subcortical, default-mode, salience, and visual subnetworks with other subnetworks is a hallmark of successful memory encoding. Moreover, multivariate analysis using the graph metrics of integration reliably classified the brain network states into the period of high (vs. low) memory encoding performance. Our findings suggest that a diverse set of brain systems dynamically interact to support successful memory encoding.

## Introduction

In everyday life, new memories of events and episodes are constantly formed, sometimes incidentally. For instance, even when we do not explicitly try to memorize a scene, we are often able to vividly recall it later. The mechanisms underlying memory encoding have extensively been investigated in cognitive psychology and neuroscience. Evidence from neuroimaging studies has shown that activation/deactivation of specific sets of brain regions is predictive of successful memory encoding (*Buckner and Wheeler, 2001*; *Fernández and Tendolkar, 2001*; *Morcom et al., 2003*; *Simons and Spiers, 2003*; *Kao et al., 2005*; *Sommer et al., 2005*; *Uncapher and Rugg, 2009*). Studies using functional magnetic resonance imaging (fMRI) have demonstrated that regions such as the medial temporal lobes (MTL) and the prefrontal cortex show greater activation in response to stimuli successfully recalled later (vs. forgotten), a phenomenon referred to as the subsequent memory effect (SME) (*Wagner et al., 1998*; *Brewer et al., 1998*; *Paller and Wagner, 2002*; *Reber et al., 2002*; *Uncapher and Rugg, 2005*; *Kim, 2011*). On the other hand, brain regions such as the posterior cingulate cortex and temporoparietal junction tend to show stronger activation (or weaker deactivation) in response to stimuli that are later forgotten (vs. remembered), referred to as the subsequent forgetting effect (SFE) (*Wagner and Davachi, 2001*; *Otten and Rugg, 2001*; *Daselaar et al., 2004*; *Kim, 2011*). In addition, several studies have shown that successful memory encoding is related to enhanced functional connectivity (FC) between memory-related regions (*Ranganath et al., 2005*; *Summerfield et al., 2006*; *Schott et al., 2013*; *Liu et al., 2014*), such as between the hippocampus and other areas. However, memory encoding is thought to require orchestration among many brain systems beyond the so-called memory system, because encoding success depends on a range of

factors (e.g., attention, arousal, and motivation) processed in distributed brain networks (*Chun and Turk-Browne, 2007*; *Gruber et al., 2014*; *Tambini et al., 2017*). Most previous studies have examined FC from a few selected 'seed' regions, providing little evidence about how the entire brain functions as a network to support memory encoding. Therefore, the role of large-scale brain networks in memory encoding processes remains to be elucidated.

Research on large-scale functional organization in the brain has advanced substantially in the past decade (*Smith et al., 2009*; *Bullmore and Sporns, 2009*; *Power et al., 2011*; *Cole et al., 2014*; *Sporns and Betzel, 2016*; *Bassett and Sporns, 2017*). This line of research emphasizes a network view of the brain rather than local activation/deactivation, showing that patterns of FC across the brain are organized in specific ways and are relevant to behavior and cognition (*Cocchi et al., 2013*; *Cole et al., 2014*; *Shine and Poldrack, 2017*). Notably, recent studies have revealed that large-scale brain networks dynamically fluctuate, typically within a timescale of 30–40 s (*Mohr et al., 2016*; *Wang et al., 2016*; *Braun et al., 2015*). Furthermore, these studies have shown that dynamic fluctuations of large-scale FC patterns are associated with a variety of cognitive processes (*Bassett et al., 2011*; *Cole et al., 2014*; *Bassett et al., 2015*; *Sadaghiani et al., 2015*; *Wang et al., 2016*; *Mohr et al., 2016*; *Cohen, 2017*; *Kucyi et al., 2018*), and even exist during the resting state (*Zalesky et al., 2014*; *Calhoun et al., 2014*; *Allen et al., 2014*; *Betzel et al., 2016*; *Shine et al., 2016*). These findings have spurred emerging perspectives of dynamic brain networks, leading researchers to focus more on time-varying FC patterns, instead of traditional 'static' FC computed in periods of 6–10 min.

Integration and segregation are key concepts in characterizing dynamic brain networks (*Friston, 2009*; *Sporns, 2013*; *Shine and Poldrack, 2017*). Theoretically, integration of large-scale networks is important for efficient communication across entire systems, whereas segregation is critical for specialized functioning of particular modules without interference from the rest of the network (*Bassett et al., 2013*; *Bassett et al., 2015*; *Sadaghiani et al., 2015*). Accumulating evidence suggests that the degrees of integration and segregation in the brain dynamically change over time (*Deco et al., 2015*; *Cohen and D'Esposito, 2016*; *Shine et al., 2016*; *Lord et al., 2017*). For example, when the brain processes a cognitively demanding task (e.g., the N-back working memory task), the degree of integration tends to increase, which is suitable for efficient communication among the sensory, motor and cognitive control systems. On the other hand, the degree of segregation tends to increase over time as the brain learns specialized skills, which allows automatic processing of a habitual task without effortful cognitive control (*Bassett et al., 2015*; *Mohr et al., 2016*). Together, it is likely that the brain changes its large-scale network configurations (i.e., integration and segregation) in highly adaptive ways. However, research on dynamic reconfigurations of large-scale brain networks is still nascent, and the findings so far suggest that the relative importance between integration and segregation is strongly dependent on tasks and situations (*Cohen and D'Esposito, 2016*; *Gonzalez-Castillo et al., 2015*), making it difficult to draw comprehensive conclusions at this stage. Thus, it remains an open question whether integration or segregation is important for memory encoding processes.

In the present study, we examined whether and how dynamic FC patterns in the brain are related to memory encoding with distinct but complementary aims. The first aim was to clarify dynamic functional connectivity patterns in well-established memory-related regions. For this, we constructed a network consisting of the brain regions associated with the SME and those associated with the SFE (hereafter referred to as the SME/SFE regions, for simplicity). Capitalizing on previous research suggesting functional interactions among SME regions for successful memory encoding (*Kim, 2011*), we predicted that the SME regions would show greater FC during time periods of higher (vs. lower) memory encoding performance. The second aim was to explore whether and how dynamic fluctuations in large-scale networks across the brain are related to memory encoding performance. For this analysis, we used a functional atlas consisting of 224 nodes that cover the entire brain (*Power et al., 2011*). Using graph theory analysis (*Rubinov and Sporns, 2010*), we quantified the degrees of integration and segregation in the large-scale network, and tested whether these graph metrics differed between the time periods of high encoding performance and those of low encoding performance.

# Results

We used an incidental memory task (*Wagner et al., 1998*; *Paller et al., 1987*), in which participants (n = 25) in a MRI scanner were presented with pictorial stimuli and instructed to make a semantic judgment about the content of each image (man-made or natural), without knowing about a subsequent surprise memory test (*Figure 1A*). To investigate dynamic fluctuations in FC patterns associated with memory encoding performance, we examined time-varying FC within a period of 36 s (consisting of 50 time points, given our sampling rate of 0.72 s; *Figure 1B,C*).

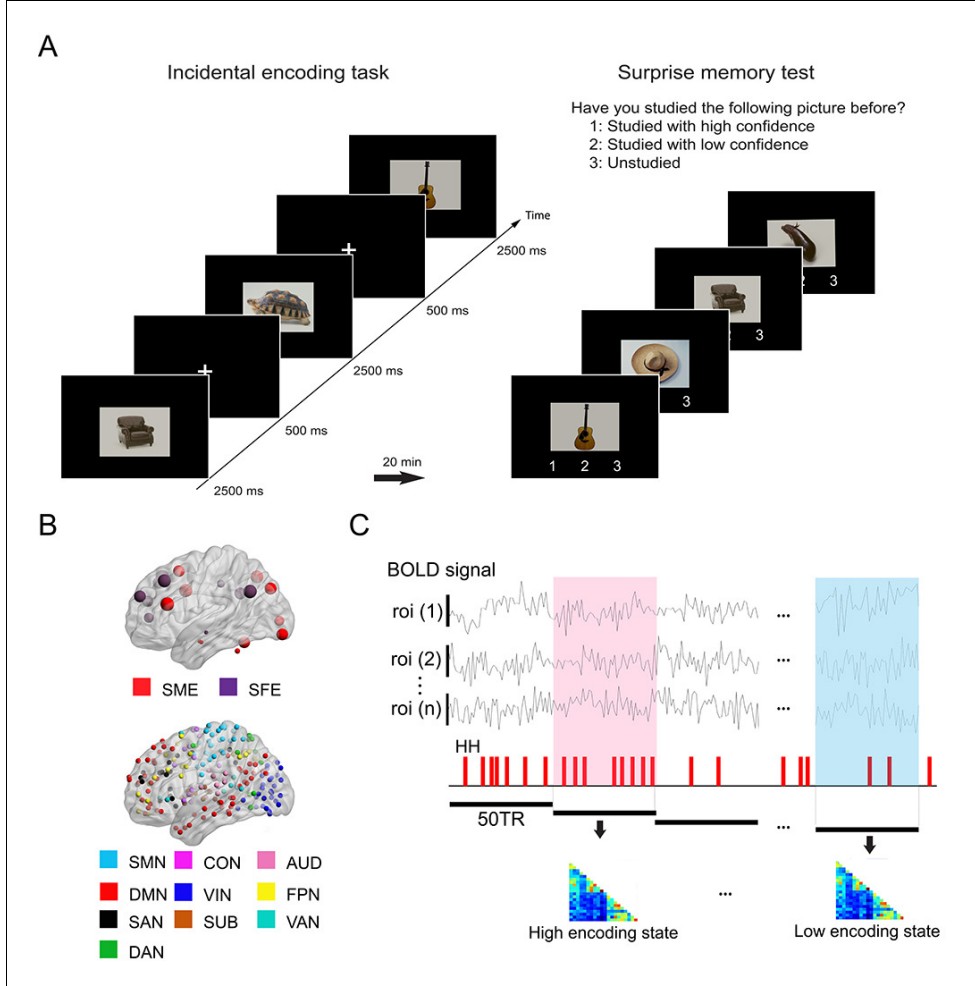

**Figure 1.** Experimental design and analysis overview. (**A**) Participants performed an incidental memory encoding task inside the scanner (360 picture trials). They judged whether each picture contained a man-made or natural object. Twenty minutes later, participants performed a surprise memory test outside the scanner (360 studied and 360 unstudied pictures). They were asked to answer if they recognized each picture as (1) studied with high confidence, (2) studied with low confidence, or (3) unstudied. (**B**) Regions of interest (ROIs) used in connectivity analysis. We used two sets of ROIs: One consisting of 21 well-established memory-related brain regions derived from a recent meta-analysis (*Kim, 2011*) and the other consisting of 224 ROIs across the whole brain derived from a functional atlas (*Power et al., 2011*). (**C**) fMRI signal time series was extracted from each ROI and divided into 36 s time windows. Each window was classified as high or low encoding state based on window-wise encoding performance, that is, the proportion of high-confidence hit (HH) trials during that time window. Functional connectivity patterns and graph metrics were estimated within each window, then averaged within each state. SME, subsequent memory effect; SFE, subsequent forgotten effect; SMN, sensorimotor networks; CON, cingulo-opercular network; AUD, auditory network; DMN, default mode network; VIN, visual network; FPN, fronto-parietal network; SAN, salience network; SUB, subcortical network; VAN, ventral attention network; DAN, dorsal attention network.

## Behavioral results

Although participants were not informed about the surprise memory test after the fMRI scan, they were able to correctly distinguish between studied and unstudied pictures with accuracy of 74.2 ± 6.3% (mean ± SD across participants). 67.7 ± 15.9% of the studied pictures were judged as studied (i.e., hit), whereas 80.8 ± 14.6% of the unstudied pictures were judged as unstudied (i.e., correct rejection). Based on the individual participants' responses in the surprise memory test, the picture trials of the incidental encoding task were categorized into high-confidence hit (HH, the pictures later remembered with high confidence; 48.9 ± 15.4%), low-confidence hit (LH, the pictures later remembered with low confidence; 18.8 ± 8.9%), or Miss (the picture later forgotten; 32.3 ± 15.9%) trials.

## Classification of time windows based on encoding performance

To relate dynamic FC patterns to incidental memory encoding performance, we divided the fMRI time series into non-overlapping small time windows (each consisting of 50 repetition time (TR) or 36 s; 45 windows per participant), and classified them into two groups based on encoding performance defined for each window. Specifically, we first defined window-wise encoding performance by computing the proportion of HH trials (the number of HH trials divided by the number of picture trials) for each time window. We then classified the time windows into either high or low encoding states on the basis of the window-wise encoding performance, with median split at participant-specific cut-off points (see Materials and methods for details). *Figure 2* shows the distributions of the windows as a function of the window-wise encoding performance, pooled across all participants. We confirmed that the number of windows classified as the high encoding state (22.7 ± 1.9, mean ± SD

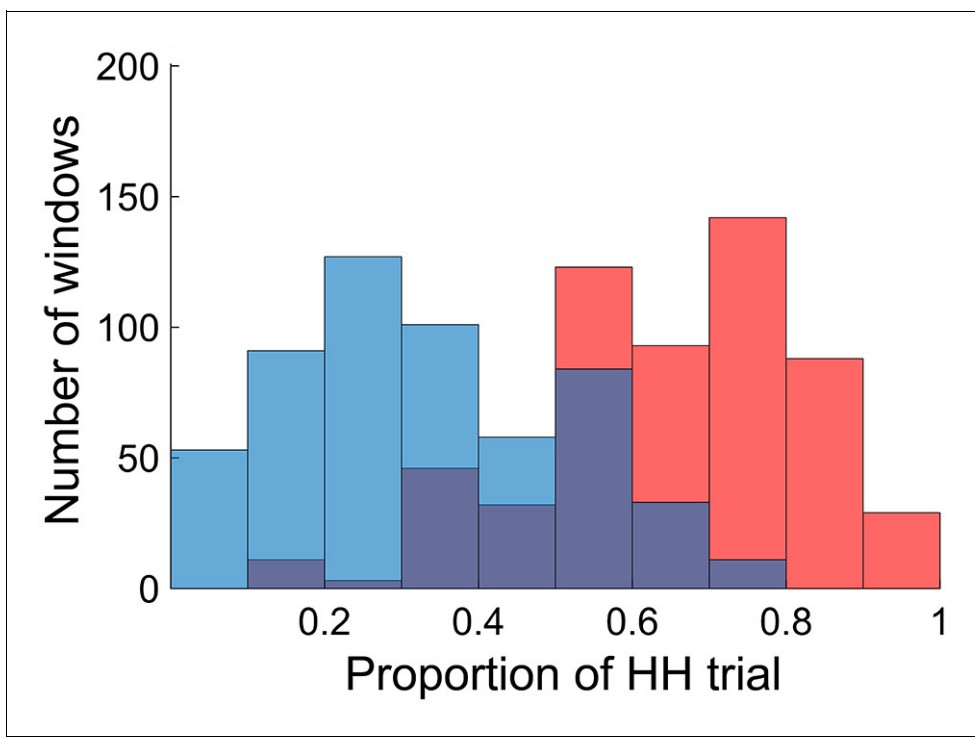

**Figure 2.** Distributions of time windows classified as high and low encoding states. The histogram shows distributions of the time windows with regard to the window-wise encoding performance, pooled across participants (red, high encoding state; blue, low encoding state). Note that the two distributions are overlapping because we used participant-specific median split to classify the high and low encoding states. Source data for the analyses reported here are available in *Figure 2—source data 1*.

The online version of this article includes the following source data for figure 2:

**Source data 1.** Proportion of HH trials of all windows, high encoding windows, and low encoding windows pooled across 25 participants.

across participants) and the low encoding state (22.3 ± 1.9) were closely matched (Wilcoxon signed-rank test, $z_{24)}$=1.0968, p=0.2727). The proportion of HH trials was 64.6 ± 15.4% (5.18 ± 1.25 trials per window) for the high encoding state and 32.6 ± 15.0% (2.59 ± 1.17 trials per window) for the low encoding state, confirming a difference in encoding performance between the two states.

It should be noted that the proportion of the high encoding state was close to 50% in all three sessions (session 1: 51.2 ± 19.7%, session 2: 45.9 ± 18.7%, session 3: 54.1 ± 13.2%, mean ± SD across participants), with no increasing or decreasing trend over time ($F_{(2,24)}$ = 1.020, p=0.367, one-way ANOVA). This ruled out the possibility that window-wise encoding performance is influenced by a mere effect of temporal proximity to the surprise memory test. In addition, we computed the proba-bility of 'state switching' (i.e., a window followed by the other type of window, such as high to low or low to high). If the state of each window was random and independent of the previous state, the probability of state switching would be approximately 50%. However, we found that the probability was significantly lower than the theoretical chance level of 50% (41.0%, p<0.001, permutation test). This indicated a history dependence of window-wide encoding performance, such that the state type of a window tended to be carried over to the next window.

## FC patterns among memory encoding-related regions

Do FC patterns differ between the high and low encoding states? To examine this issue, we first focused on well-established memory encoding-related brain regions. Based on a recent meta-analy-sis (*Kim, 2011*), we defined a brain network consisting of 11 SME regions and 10 SFE regions (*Figure 1B*; for the detail of the regions of interest (ROIs), see *Supplementary file 1A*). By focusing on functionally well-characterized regions, we aimed to test our hypothesis regarding dynamic FC patterns. Specifically, we predicted greater FC among the SME regions in the high encoding state relative to the low encoding state, given the proposed functional interactions among the SME regions in successful memory encoding (*Kim, 2011*). Importantly, trial-related activation analysis of our own fMRI data confirmed the SME and SFE in these ROIs (*Figure 3A–C*; *Supplementary file 1B*).

To examine FC patterns, we calculated Fisher's Z-transform of Pearson's correlation coefficients of the windowed time series between all pairs of ROIs. The connectivity matrices were then aver-aged across time windows separately for the high and low encoding states (*Figure 3D and E*). While the connectivity matrices of the two states appeared to be similar, direct comparison revealed a trend indicating FC increases during the high vs. low encoding state between some pairs of SME-related ROIs, most notably in the right middle occipital gyrus (MOG)-right hippocampus/parahippo-campal gyrus (HCP/PHG) pair, the right MOG-left fusiform gyrus pair, and the left inferior occipital gyrus-left HCP/PHG pair (Wilcoxon signed-rank test, $z_{(24)}$ = 2.3274–3.0808, p=0.0021–0.0199; *Figure 3F*). However, it should be noted that these results did not survive false-discovery-rate (FDR) correction among $_{21}C_2$ = 210 tests. We also computed 'within-subnetwork' connectivity by averaging the values in the connectivity matrices among the SME regions and among the SFE regions, respec-tively. Neither the within-subnetwork connectivity for the SME regions nor that for the SFE regions showed a significant difference between the high and low encoding states (Wilcoxon signed-rank test; SME: $z_{(24)}$ = 0.4709, p=0.6377; SFE: $z_{(24)}$ = 0.6592, p=0.5098; *Figure 3G*). A two-way ANOVA testing the interaction between region (SME vs. SFE regions) and state (high vs. low encoding states) did not reveal a significant interaction ($F_{(1,24)}$=0.1298, p=0.7218). Overall, although we observed a trend that was consistent with our prediction, the results were not statistically significant. This finding may imply that dynamic FC patterns associated with memory encoding performance cannot be effectively captured if by examining only the 'memory-related' brain regions identified in previous activation-based research (see Discussion for more details).

## FC patterns across large-scale brain networks

Next, to elucidate how a diverse set of brain systems are coordinated for successful memory encod-ing, we examined FC patterns across a large-scale brain network. In this analysis, we defined a brain-wide network consisting of 224 ROIs (organized into 10 subnetworks; *Figure 1B*; *Supplementary file 1C*). This network was derived from a well-established functional brain atlas (*Power et al., 2011*), and the same ROIs and subnetwork labels have been used in many previous studies investigating dynamic/static FC during task fMRI (*Cole et al., 2014*; *Cohen et al., 2014*;

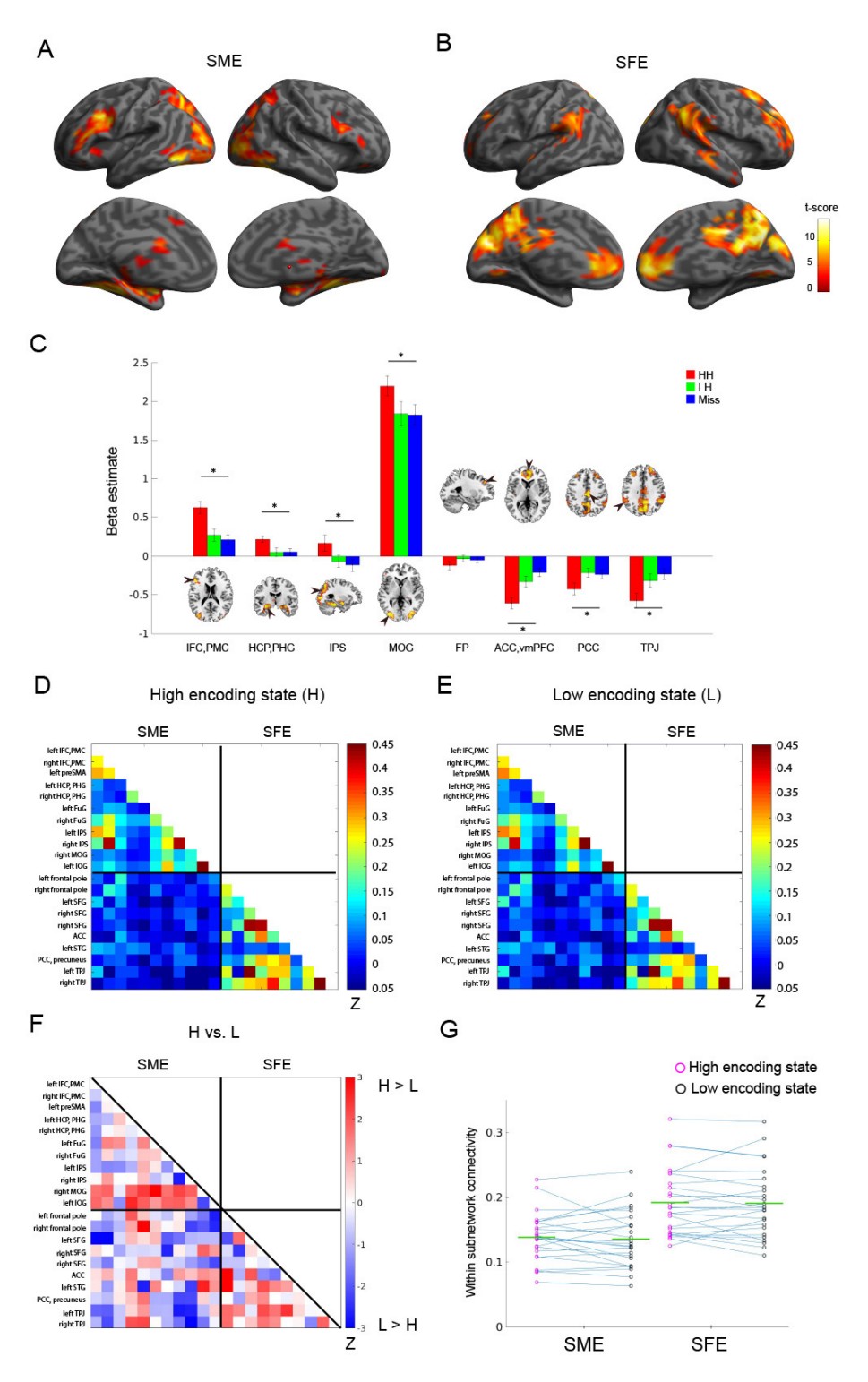

**Figure 3.** Functional connectivity patterns among memory-related brain regions. Trial-related activation analysis confirmed (**A**) the subsequent memory effect (SME, that is, HH >Miss) and (**B**) subsequent forgetting effect (SFE, that is, Miss > HH). Statistical parametric maps are thresholded at p<0.05, FWE corrected across the whole brain. (**C**) Bar graph shows beta estimates (mean ± SEM across participants) for high-confidence hit (HH), low-confidence hit (LH), and Miss trials in representative ROIs from the SME/SFE regions. (**D**) Connectivity matrix of the high encoding state, averaged across participants. (**E**) Connectivity matrix of the low encoding state, averaged across

*Figure 3 continued on next page*

*Figure 3 continued*

participants. Color bars indicate Fisher-Z transform of Pearson's correlation coefficients. (F) A matrix illustrating statistical differences in functional connectivity patterns between the high and low encoding states. Color bar indicates z values derived from Wilcoxon signed-rank test across participants. Connections showing significant differences (p<0.05, FDR corrected) are marked in red in the upper triangle of the matrix. (G) Within-subnetwork connectivity for the SME and SFE regions (mean Fisher's Z value of connections within the SME and SFE regions, respectively). Magenta and black circles represent individual-participant data for the high and low encoding states, respectively. Green horizontal lines indicate across-participant means. Asterisk indicates a significant difference in within-subnetwork connectivity between the high and low encoding states (Wilcoxon signed-rank test, p<0.05). IFC, inferior frontal cortex; PMC, premotor cortex; HCP, hippocampus; PHG, parahippocampal gyrus; IPS, intraparietal sulcus; MOG, middle occipital gyrus; FP, frontal pole; ACC, anterior cingulate cortex; vmPFC, ventromedial prefrontal cortex; PCC, posterior cingulate cortex; TPJ, temporoparietal junction. Source data for the analyses reported here are available in *Figure 3—source data 1–5*.

The online version of this article includes the following source data for figure 3:

**Source data 1.** Beta estimates extracted from 8 representative ROIs from the SME/SFE regions.
**Source data 2.** Mean correlation coefficients across the 21 SME/SFE regions, averaged across high encoding windows.
**Source data 3.** Mean correlation coefficients across the 21 SME/SFE regions, averaged across low encoding windows.
**Source data 4.** z-values derived from Wilcoxon signed-rank test across participants.
**Source data 5.** Mean Fisher's z-value of connections within the SME and SFE regions for each participant, separately for the high and low encoding states.

---

*Braun et al., 2015*; *Cohen and D'Esposito, 2016*; *Mohr et al., 2016*; *Westphal et al., 2017*). To obtain FC patterns, we calculated pairwise correlations of the windowed time series among the 224 ROIs (*Figure 4A and B*), just as we did for the SME/SFE networks. By comparing the high and low encoding states, we found significant differences in FC associated with encoding performance: 98 connections showed significant increases in FC during the high encoding states, whereas 687 connections showed significant decreases (surviving FDR corrections among $_{224}C_2$ = 24,976 tests; *Figure 4C*; *Supplementary file 1D and 1E*). Three-dimensional (3D) visualizations of differential FC patterns are shown in *Figure 4D and E*. Interestingly, the connections showing significant increases in FC during the high encoding state tended to be long range (Euclidean distance: 84.6 ± 27.3; *Figure 4D*), whereas those showing significant decreases tended to be short range (Euclidean distance: 76.5 ± 29.7; *Figure 4E*; increases vs. decreases: $z_{(783)}$ = 2.7846, p<0.0054, Wilcoxon rank-sum test). These observations suggest a systematic reconfiguration of the large-scale network between the high and low encoding states, rather than a homogeneous, brain-wide increase or decrease in FC.

In light of the pivotal role of the hippocampus in memory encoding, we also examined the FC patterns between the hippocampus and the large-scale network. In this analysis, we considered a 226-node network that combined the bilateral hippocampus ROIs (*Kim, 2011*) with the 224 ROIs (*Power et al., 2011*). As expected, we observed increased FC between the hippocampus and occipital cortex during the high vs. low encoding state (right hippocampus-left MOG: $z_{(24)}$ = 3.5383, p<0.004; surviving FDR corrections among $_{226}C_2$ = 25,425 tests; *Figure 4—figure supplement 1*). The left hippocampus showed increased FC with the right thalamus ($z_{(24)}$ = 3.2157, p=0.0013; surviving FDR corrections).

## Graph analysis on large-scale brain network

The results described above imply a dynamic reconfiguration of a large-scale brain network between the high and low encoding states. In particular, the high encoding state appears to be characterized by enhanced long-range FC among distant brain regions, whereas the low encoding state seems to be characterized by increased local connectivity among neighboring regions. This may indicate that the brain shows different levels of functional integration/segregation depending on encoding performance. To formally test this possibility, we applied graph theory to derive measures of integration and segregation from the 224-node network. First, we computed global efficiency, a measure of integration defined for the entire network, and local efficiency (averaged across all nodes), a measure of segregation. We found that global efficiency was significantly higher during the high (vs. low)

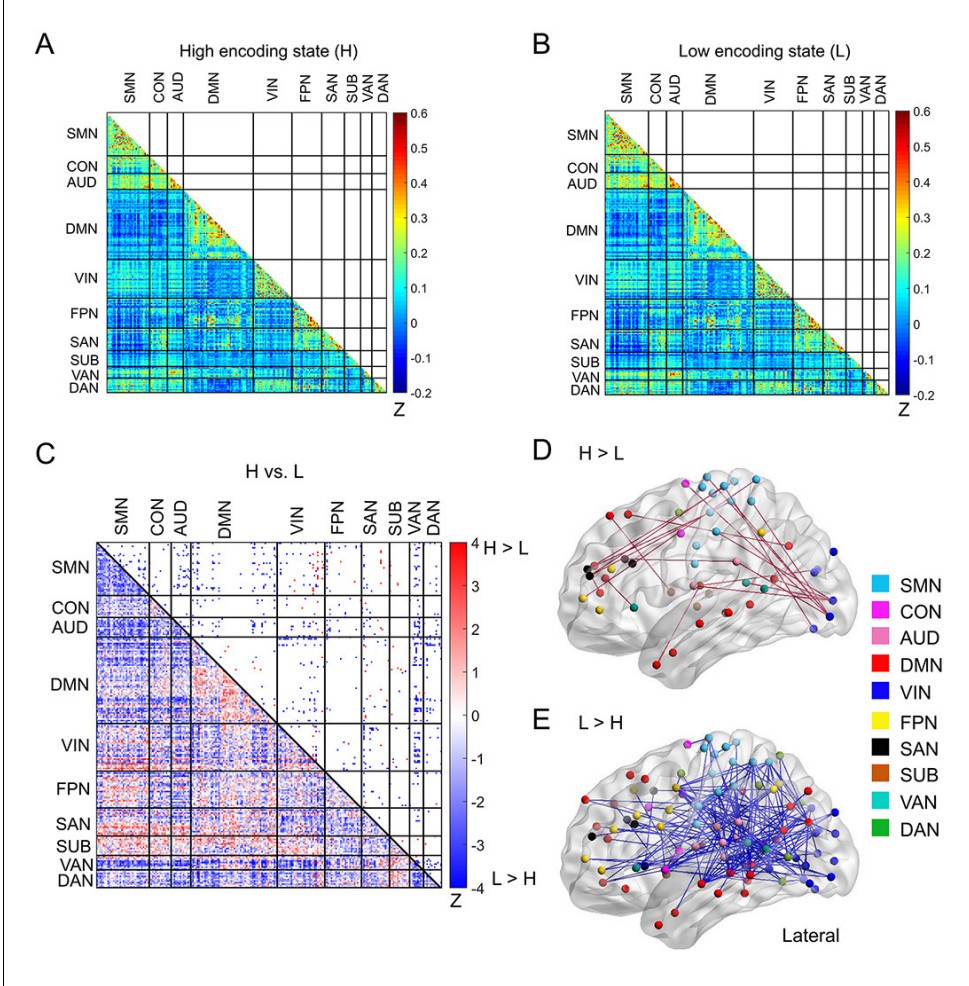

**Figure 4.** Functional connectivity patterns across large-scale brain network. (**A**) Connectivity matrix of the high encoding state, averaged across participants. (**B**) Connectivity matrix of the low encoding state, averaged across participants. Color bars indicate Fisher's Z-transform of Pearson's correlation coefficients. (**C**) A matrix illustrating statistical differences in functional connectivity patterns between the high and low encoding states. ROIs belonging to the same subnetwork were grouped together resulting in 10 subnetworks. Color bar indicates z values derived from Wilcoxon signed-rank test across participants. Connections showing significant differences (p<0.05, FDR corrected) are marked in color (red, high >low; blue, low >high) in the upper triangle of the matrix. (**D**) Three-dimensional (3D) visualizations of significantly greater functional connectivity during the high encoding state. (**E**) 3D visualizations of significantly greater functional connectivity during the low encoding state. Source data for the analyses reported here are available in *Figure 4—source data 1–5*.

The online version of this article includes the following source data and figure supplement(s) for figure 4:

**Source data 1.** Mean correlation coefficients across the 224 ROIs, averaged across high encoding windows.
**Source data 2.** Mean correlation coefficients across the 224 ROIs, averaged across low encoding windows.
**Source data 3.** z-values derived from Wilcoxon signed-rank test across participants.
**Source data 4.** .edge file used as an input of BrainNet Viewer toolbox to create the 3D visualizations of significantly greater functional connectivity during the high encoding state shown in *Figure 4D*.
**Source data 5.** .edge file used as an input of BrainNet Viewer toolbox to create the 3D visualizations of significantly greater functional connectivity during the low encoding state shown in *Figure 4E*.
**Figure supplement 1.** Difference in functional connectivity patterns between the high and low encoding states in a 226-node network combining the bilateral hippocampus and the Power atlas.
**Figure supplement 2.** Functional connectivity patterns across a 285-node network derived from the Gordon atlas.

encoding state ($z_{(24)}$ = 3.9688, p=0.0001; *Figure 5A*), whereas local efficiency was not different between the states ($z_{(24)}$ = 0.6861, p=0.4926; *Figure 5C*). A two-way ANOVA showed a significant interaction ($F_{(1,24)}$ = 17.1006, p=0.0004) between state (high vs. low) and metrics (global vs. local efficiency), confirming that only the measure of integration changed in association with memory encoding performance. Next, to analyze the network architecture in more detail, we examined subnetwork-wise measures of integration and segregation. Specifically, we computed participation coefficients (PCs) and local efficiency averaged across nodes within each subnetwork, as measures of integration and segregation, respectively (*Power et al., 2013*; *Marek et al., 2015*; *Shine et al., 2016*). For subnetwork-wise PCs, we observed a significant state-by-subnetwork interaction ($F_{(9,216)}$ = 4.4700, p=2.1613 × $10^{-5}$; two-way ANOVA), indicating that the difference in PCs between the

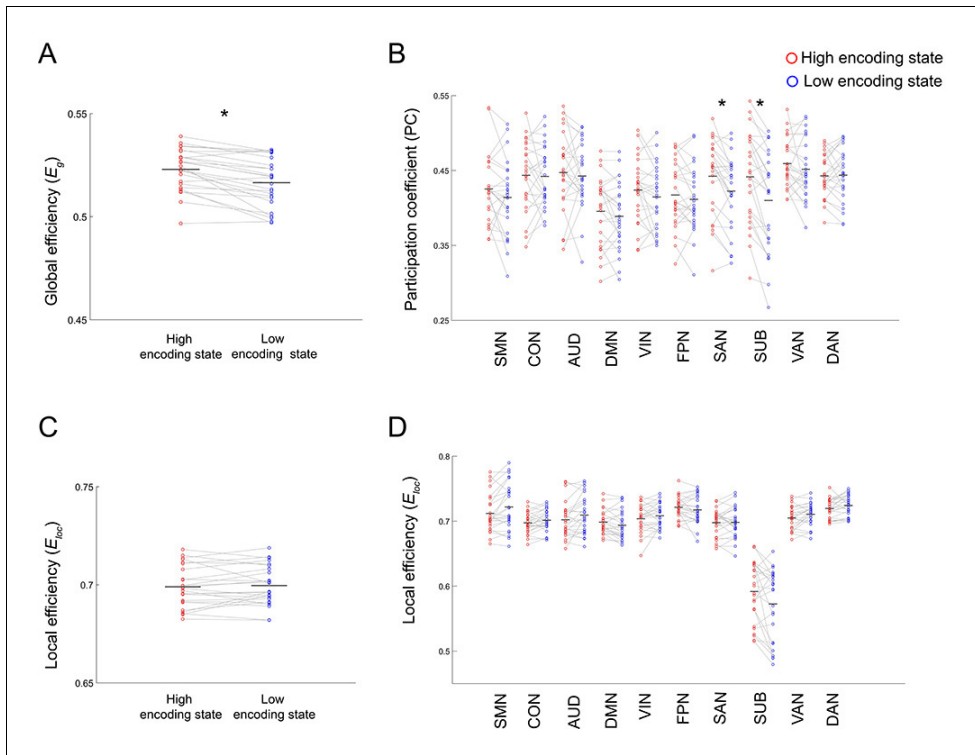

**Figure 5.** Differences in integration/segregation of large-scale network between high and low encoding states. (A) Global efficiency, a measure of network integration. (B) Participation coefficient (averaged across nodes within each subnetwork), a measure of integration defined at subnetwork level. (C) Local efficiency (averaged across all nodes), a measure of segregation. (D) Local efficiency (averaged across nodes within each subnetwork), a measure of segregation defined at the subnetwork level. Graph metrics were computed for each time window, then averaged across windows separately for each state. Dots represent individual-participant data. Black horizontal lines indicate across-participant means. Asterisks indicate a significant difference between the states (p<0.05, FDR corrected for subnetwork-wise metrics). Source data for the analyses reported here are available in *Figure 5— source data 1–4*.

The online version of this article includes the following source data and figure supplement(s) for figure 5:

**Source data 1.** Mat file containing global efficiency in the high and low encoding states for each participant.
**Source data 2.** Participation coefficients (averaged across nodes within each subnetwork) in the high and low encoding states for each participant.
**Source data 3.** Local efficiency (averaged across all nodes) in the high and low encoding states for each participant.
**Source data 4.** Local efficiency (averaged across nodes within each subnetwork) in high and low encoding states for each participant.
**Figure supplement 1.** Levels of integration/segregation of a 224-node network associated with encoding performance.
**Figure supplement 2.** Differences in integration/segregation between high and low encoding states for a 285-node network derived from the Gordon atlas.
**Figure supplement 3.** Edge reliability analysis.

high and low encoding states varied across the subnetworks. When we examined individual subnetworks, we found significantly higher PCs during the high (vs. low) encoding state in two subnetworks (salience network [SAN]: $z_{(24)}$ = 3.0001, p=0.0027; subcortical nodes [SUB]: $z_{(24)}$ = 3.7535, p=0.0002; surviving FDR correction among 10 tests; *Figure 5B*). In addition, we observed a significant state-by-subnetwork interaction ($F_{(9,216)}$ = 5.4792, p=8.7644 $\times$ $10^{-7}$; two-way ANOVA) for subnetwork-wise local efficiency. However, when we examined each subnetwork separately, we did not find a significant difference between the states in any specific subnetwork after FDR correction ($z_{(24)}$ < |2.4351|, p>0.0149; *Figure 5D*). To clarify the differential results between the subnetwork-wise measures of integration and segregation, we performed a two-way ANOVA testing the interaction between state (high vs. low) and metric (subnetwork-wise PC vs. subnetwork-wise local efficiency) for each subnetwork separately. We found a significant state-by-metric interaction only in the SAN ($F_{(1,24)}$ = 11.9399, p=0.0021; surviving FDR correction among 10 tests). This result confirmed that only the subnetwork-wise integration, not segregation, was associated with memory encoding performance in the SAN.

Furthermore, we asked whether functional integration between specific pairs of subnetworks differs between the high and low encoding states. For this aim, we defined 'inter-subnetwork efficiency' ($E_{is}$), which quantifies integration between each subnetwork pair. We found significant differences in inter-subnetwork efficiency between the high and low encoding states in many subnetwork pairs (surviving FDR correction among $_{10}C_2$ = 45 tests; *Figure 6*). Specifically, higher inter-subnetwork efficiency for the high encoding state was observed among some subnetworks, e.g., between the SUB, SAN, DMN, and VIN and the rest of the subnetworks ($z_{(24)}$ > 2.3274, p<0.0199). Only the SMN-AUD pair showed lower inter-subnetwork efficiency for the high encoding state ($z_{(24)}$ = −2.7041, p=0.0068).

We also examined modularity, a measure of entire network-level segregation indicating how well a network can be partitioned into distinct communities (*Westphal et al., 2017*; *Wig, 2017*). Unlike the results for local efficiency, we observed significantly higher modularity during the high encoding state ($z_{(24)}$ = 2.5427, p=0.0110). Given that global efficiency (a measure of integration) was higher during the high encoding state, one may expect to observe lower modularity in the high encoding state. In theory, however, a network can exhibit both high integration and high modularity simultaneously (*Pan and Sinha, 2009*; *Meunier et al., 2010*). To clarify the relationship between modularity and global efficiency, we computed a window-to-window correlation (within each individual) between the two metrics. We found no significant correlation between the two metrics (Pearson's $r$ = 0.1155 ± 0.4408; $z_{(24)}$ = 1.3319, p=0.1829), suggesting that these two graph metrics captured somewhat independent aspects of the network architecture. We also computed a window-to-window correlation between modularity and entire-network local efficiency, revealing a positive correlation (Pearson's $r$ = 0.5907 ± 0.1525; $z_{(24)}$ = 4.3724, p=1.2290 $\times$ $10^{-5}$), consistent with the notion that both modularity and local efficiency are metrics of segregation.

We also examined the modularity contribution of each subnetwork (see Materials and methods). We found a significant state-by-subnetwork interaction ($F_{(9,\ 216)}$=3.4900, p=0.0005, two-way ANOVA), revealing that the modularity-contribution differences between the states varied across the subnetworks. At the subnetwork level, we found trends indicating greater modularity contribution during the high vs. low encoding state in three subnetworks, most notably in the SUB (SUB: $z_{(24)}$ = 2.7311, p=0.0063; DAN: $z_{(24)}$ = 2.3274, p=0.0199; DMN: $z_{(24)}$ = 2.0584, p=0.0396; Wilcoxon signed-rank test; see *Supplementary file 1F*), although these results did not survive FDR correction. To further clarify the trends of increased modularity contributions (high vs. low encoding state) in these subnetworks, we examined the proportion of edges within single subnetworks and those connecting different subnetworks (*Supplementary file 1F*). We found that both the proportion of edges within the SUB and those across the SUB and the other subnetworks increased during the high encoding state (within: $z_{(24)}$ = 3.0003, p=0.0027; across: $z_{(24)}$ = 2.7580, p=0.0058). Similar patterns were not observed in the DAN or DMN. In addition, the increase in the proportion of edges (high vs. low encoding states) was greater for the connections within the SUB than those across the SUB and the other subnetworks ($z_{(24)}$ = 2.5965, p=0.0094), in line with the increased modularity contribution observed in the SUB. A two-way ANOVA revealed an interaction ($F_{(1,\ 24)}$=8.4671, p=0.0077) between state (high vs. low) and connection type (within vs. across). These findings suggest that the SUB contributed to entire network-level segregation (i.e., modularity) by its increased within-subnetwork connections, and also contributed to inter-subnetwork integration by its increased connection with other subnetworks.

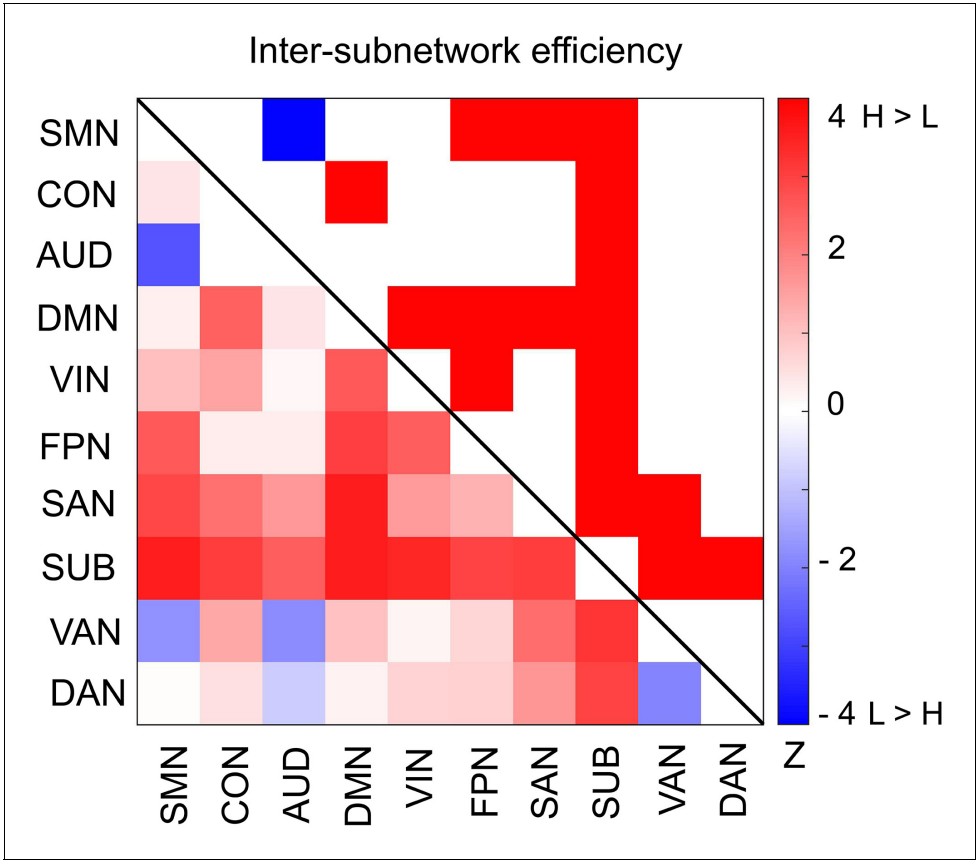

**Figure 6.** Differences in inter-subnetwork efficiency between high and low encoding states. Inter-subnetwork efficiency ($E_{is}$) was computed for each pair of subnetworks. Color bar indicates z values derived from Wilcoxon signed-rank test across participants. Subnetwork pairs showing significant differences (p<0.05, FDR corrected) are marked in color (red, high >low; blue, low >high) in the upper triangle of the matrix. Source data for the analyses reported here are available in *Figure 6—source data 1*.

The online version of this article includes the following source data and figure supplement(s) for figure 6:

**Source data 1.** z-values derived from Wilcoxon signed-rank test across participants.

**Figure supplement 1.** Differences in inter-subnetwork efficiency between high and low encoding states for a 285-node network derived from the Gordon atlas.

A previous study reported that the DMN can be split into submodules during performance of a memory recollection task (*Fornito et al., 2012*). To determine whether submodule structure within the DMN is also relevant to memory encoding, we tested whether the DMN could be divided into submodules during the high and low encoding states, and whether the DMN submodules showed distinct profiles of 'submodule-wise' integration and segregation. By applying a group-level modular decomposition method (*Fornito et al., 2012*), we found that the DMN (consisting of 56 nodes) was divided into five submodules (*Figure 7*; *Supplementary file 1G*). Importantly, the submodule assignment of the DMN nodes was identical between the high and low encoding states, suggesting that the DMN submodule structure did not change between the two states. Next, we computed the submodule-wise PC and local efficiency by averaging the node-wise metrics within each of the five DMN submodules. We found no significant difference in the submodule-wise PC ($|z_{(24)}|<1.6279$, p>0.1036). We observed a trend indicating higher submodule-wise local efficiency in one submodule (submodule 3, consisting of nodes in the medial PFC; $z_{(24)} = 2.3274$, p=0.0199), but this was not significant after FDR correction among five tests, and the state-by-submodule interaction was not significant ($F_{(4, 96)}=1.8932$, p=0.1179; two-way ANOVA). Although it is possible that segregation in a specific DMN submodule is related to a better encoding performance, further research is needed to elucidate the possible differentiation among the DMN submodules for memory encoding.

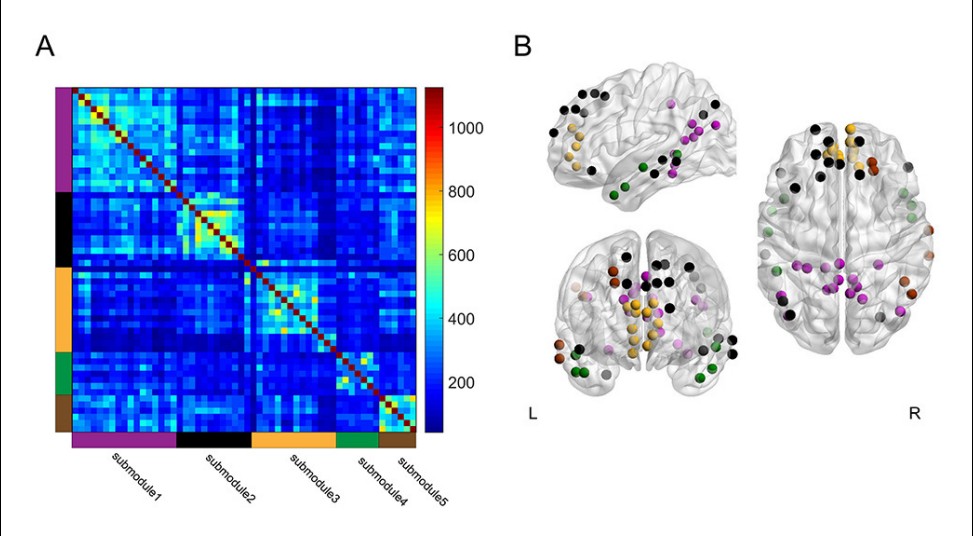

**Figure 7.** Group-level module decomposition of the DMN subnetwork. (**A**) Group consistency coclassification matrix indicating an optimal modular structure across all time windows and participants. (**B**) 3D visualization of the 56 DMN nodes classified into five submodules. Source data for the analyses reported here are available in *Figure 7—source data 1* and *2*.

The online version of this article includes the following source data for figure 7:

**Source data 1.** The matrix showing how frequently each node was assigned to be the same submodule.
**Source data 2.** .node file used as an input of BrainNet Viewer toolbox to create the 3D visualizations of the 56 DMN nodes classified into the five submodules as shown in *Figure 7B*.

## Multivariate pattern classification using graph metrics as features

Graph analysis is considered an effective method for extracting a concise set of features from a large-scale network. If a set of graph metrics (e.g., participation coefficients computed at a subnetwork level) represent the large-scale network architecture well, one network state can be discriminated from another using multi-dimensional vectors of the graph metrics, instead of using the entire connectivity matrices. Building on this idea, we attempted to classify the high and low encoding states using the graph metrics of integration and segregation derived from the 224-node network. Specifically, we performed across-participant binary classification using support vector machine (SVM) with leave-one-out cross validation. When we used the subnetwork-wise PCs (i.e., 10 features) as the input of the SVM classifier, we were able to reliably distinguish the high from low encoding states with 74% classification accuracy (p=0.0094, permutation test; *Figure 8A*). Likewise, when we used subnetwork-wise local efficiency, we obtained a classification accuracy of 68% (p=0.0267, permutation test; *Figure 8B*). Using inter-subnetwork efficiency (i.e., 45 features) as inputs, we obtained a classification accuracy of 72% (p=0.0029, permutation test; *Figure 8C*). Notably, when we used the entire FC patterns (i.e., Fisher Z-transform of Pearson's correlation coefficients, $_{224}C_2$ = 24,976 features) as the input, classification accuracy dropped to chance levels (36%, p=0.9406, permutation test), likely because of the curse of dimensionality.

To ensure that the observed differences in the functional network architecture associated with encoding performance were not caused by other confounding factors, such as visual responses to pictures or reaction times (RTs) for semantic judgment, we performed two control analyses, as follows. First, to assess the influence of simple visual stimulation on functional network architecture, we sorted the windows based on the proportion of picture trials (varied across windows because of occasional fixation trials), instead of dividing the states based on encoding performance. We used the same SVM approach as above, but this time to classify the periods with more picture trials and those with fewer picture trials ('more pic' vs. 'fewer pic'; split at the participant-specific medians). In this control analysis, we did not observe significant classification accuracy with the subnetwork-wise

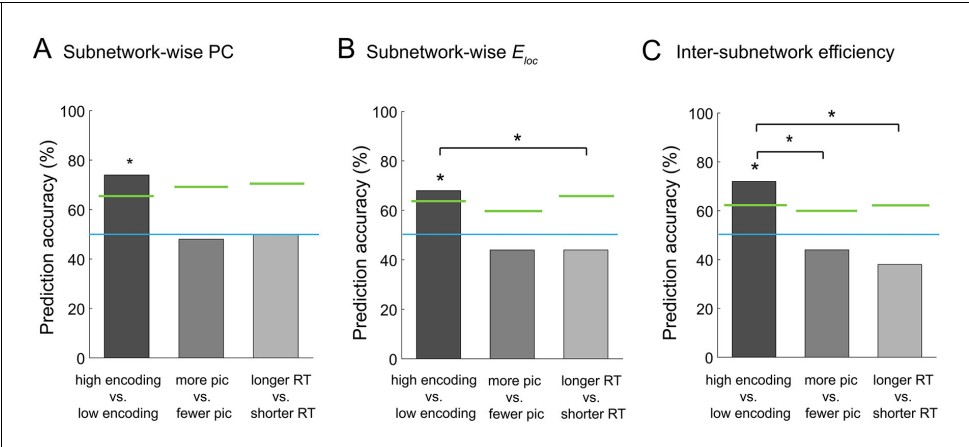

**Figure 8.** Multivariate classification analysis using graph metrics. (**A**) Prediction accuracy of support vector machine (SVM) classifier using subnetwork-wise PCs as the input. (**B**) Prediction accuracy of SVM classifier using subnetwork-wise local efficiency as the input. (**C**) Prediction accuracy of SVM classifier using inter-subnetwork efficiency as the input. Bar graphs represent prediction accuracy obtained by different criteria for sorting time windows: encoding performance and two control criteria (proportion of picture trials and RT for semantic judgment). Asterisks indicate statistical significance (p<0.05, permutation test). Green lines indicate significance threshold determined by permutation null distributions. The blue lines indicate the theoretical chance level (i.e., 50%). Source data for the analyses reported here are available in *Figure 8—source data 1*.

The online version of this article includes the following source data for figure 8:

**Source data 1.** Prediction accuracy (*P* < 0.05, permutation test), i.e., the output from the SVM classifier using subnetwork-wise PCs, subnetwork-wise local efficiency, and inter-subnetwork efficiency as inputs.

PCs (48%, p=0.6349, permutation test; *Figure 8A*), with the subnetwork-wise local efficiency (44%, p=0.8713, permutation test; *Figure 8B*), or with the inter-subnetwork efficiency (44%, p=0.8713, permutation test; *Figure 8C*). Second, to examine the influence of RT for semantic judgment, we sorted the windows based on average RT computed within each window. We ran the SVM analysis to classify the periods of longer average RT and those of shorter average RT ('longer RT' vs. 'shorter RT'; split at the participant-specific medians). Again, we did not observe significant classification accuracy with the subnetwork-wise PCs (50%, p=0.5566, permutation test; *Figure 8A*), with the subnetwork-wise local efficiency (44%, p=0.8000, permutation test; *Figure 8B*), or with the inter-subnetwork efficiency (38%, p=0.9595, permutation test; *Figure 8C*). Furthermore, we confirmed that classification based on encoding performance generally resulted in higher accuracy than that based on the proportion of pictures or RT, with PCs (encoding vs. pictures: p=0.0611; encoding vs. RT: p=0.0822; *Figure 8A*), with local efficiency (encoding vs. pictures: p=0.0721; encoding vs. RT: p=0.0311; *Figure 8B*), or with inter-subnetwork efficiency (encoding vs. pictures: p=0.0036; encoding vs. RT: p=0.0001; *Figure 8C*). Overall, these results suggest that the observed differences in functional network architecture were specifically related to encoding performance, not to simple visual stimulation or RT for semantic judgment.

## Robustness check

To check the robustness of our findings (particularly the graph analysis applied to the large-scale network; *Figure 5*), we performed a number of additional analyses. First, we confirmed that our results were robust across a range of window sizes and proportional thresholds (*Supplementary file 1H*). Importantly, our finding of higher global efficiency during the high encoding state was maintained even for 7.2 s (10 TRs) time windows. Second, the results did not change when we used overlapping sliding windows (in steps of 1 TR; *Supplementary file 1I*). Third, the results were unchanged when we shifted the time series by 5 s to take hemodynamic delay into account (*Supplementary file 1J*), or when we used task fMRI time series instead of residuals (see Materials and methods; *Supplementary file 1K*). Fourth, we confirmed that our results were unchanged when we used both high- and low-confidence hit trials to define the window-wise encoding performance (*Supplementary file 1L*). Fifth, the results remained the same when we used the top and bottom

tertiles or quartiles instead of median split to classify the time windows based on encoding performance (*Supplementary file 1M and 1N*; *Figure 5—figure supplement 1*). Finally, the results did not change after controlling for the effect of time passed within each session or across sessions (see Materials and methods; *Supplementary file 1O*), ruling out the possibility that our findings were simply driven by gradual changes in psychological states over time (e.g., a decrease in concentration/motivation) or by primacy/recency effects. It should be emphasized that the significant differences in graph metrics were observed only in association with encoding performance, not with the proportion of picture trials or with RT for semantic judgment (*Supplementary file 1P and 1Q*).

To cross-validate our findings regarding large-scale network characteristics, we repeated the same analyses using an independent atlas (*Gordon et al., 2016*), which consisted of 285 nodes (see *Supplementary file 1R*). The results were consistent across the two atlases (*Supplementary file 1S*; *Figure 4—figure supplement 2*, *Figure 5—figure supplement 2*, *Figure 6—figure supplement 1*), further demonstrating the robustness of the findings. Likewise, the graph metrics computed on the 226-node network (i.e., the bilateral hippocampus combined with the Power atlas) also showed consistent results (*Supplementary file 1T*).

## Possible effects of overall FC strength

For the graph analysis described above, we applied proportional thresholding to obtain unweighted graphs. Although this method has been widely used in previous research (*Cohen and D'Esposito, 2016*; *Sadaghiani et al., 2015*), a recent study raised a concern (*van den Heuvel et al., 2017*), suggesting a possible influence of overall FC strength on graph metrics computed by this method (it should be noted that the use of absolute thresholding or unthresholded weighted graphs may not effectively circumvent this issue, as discussed in the paper mentioned above). Specifically, if overall FC strength of a given connectivity matrix is weaker, the graph constructed from that connectivity matrix by proportional thresholding can include less-reliable (or false-positive) edges more frequently. This may confound graph metrics. In other words, differences in the reliability of edges between two graphs could result in spurious differences in graph metrics.

To refute this possibility, we examined the 'edge reliability' of large-scale graphs separately for the high and low encoding states. Here, edges were defined as reliable if they existed consistently across time windows more than by chance (see Materials and methods). We found that the proportions of reliable edges were significantly high in both high and low encoding states (relative to randomized networks; *Supplementary file 1U*), and that they were not statistically different from each other ($z_{(24)}$ = 1.2916, p=0.1965; *Figure 5—figure supplement 3*). In theory, a lower proportion of reliable edges (i.e., a higher proportion of false-positive edges) may result in a higher value of global efficiency, because it can introduce random connections between unrelated nodes (*van den Heuvel et al., 2017*). In our case, the proportion of reliable edges was numerically higher for the high encoding state across a range of proportional thresholds (*Supplementary file 1U*). Therefore, if differences in edge reliability between the two states confounded our results, we would have found higher global efficiency during the low encoding state. However, this is the opposite of what we observed, ruling out the possibility that our findings were mere artifacts arising from less reliable edges.

To further examine potential effects of overall FC strength, we performed an additional analysis using 'adjusted' graph metrics, in which we regressed out the effect of overall FC strength estimated for each time window (see Materials and methods). When we compared the high and low encoding states, we found a significant difference in modularity, but not in global efficiency or PC (see *Supplementary file 1V*). This finding suggests that overall FC strength shared a considerable amount of variance (on a window-by-window basis) with the latter three graph metrics. The multivariate analysis using the adjusted subnetwork-wise PC, subnetwork-wise local efficiency, and inter-subnetwork efficiency did not provide significant classification accuracy (62%, p=0.3532; 34%, p=0.7821; 32%, p=0.7840, permutation test, respectively). This may imply that the results for the subnetwork-wise integration and segregation associated with encoding performance are difficult to disentangle from the effect of overall FC strength, at least in this case. Because it removes potentially true information about network organization, the strategy of adjusting for overall FC strength may be overly strict, as mentioned in previous studies (*van den Heuvel et al., 2017*).

## Addressing possible concerns about motion confounds

Another issue is the potential effects of motion confounds on graph metrics. To examine this issue, we computed framewise displacement (FD) as an index of head motion (*Power et al., 2012*; *Power et al., 2014*; *Laumann et al., 2017*). We observed no significant difference in mean FD between the high and low encoding states (high: 0.1078 ± 0.0096, low: 0.1093 ± 0.0086; $z_{(24)}$ = −1.9238, p=0.0544), and there were no correlations between the window-wise FD and memory encoding performance (Pearson's $r$ = −0.0742 ± 0.1794; $z_{(24)}$ = −1.8700, p=0.0615), global efficiency (Pearson's $r$ = −0.0429 ± 0.2557; $z_{(24)}$ = −0.9014, p=0.3674), or modularity (Pearson's $r$ = 0.0160 ± 0.1872; $z_{(24)}$ = 0.1211, p=0.9036). To further ensure that our results were not derived from motion-related confounds, we performed a set of supplement analyses as below.

First, we confirmed our main results using graph measures adjusted for window-wise FD. We found that the difference in global efficiency between the high and low encoding states remained significant after we regressed out window-wise FD from global efficiency ($z_{(24)}$ = 3.9688, p=0.0002). Modularity adjusted for window-wise FD was also significantly different between the states ($z_{(24)}$ = 2.4082, p=0.0160). Furthermore, we performed multivariate classification analysis using the graph metrics adjusted for the window-wise FD. The results revealed significant classification accuracy with the subnetwork-wise PCs and inter-subnetwork efficiency adjusted for the window-wise FD (86%, p=0.0125; 72%, p=0.0034, permutation test, respectively), indicating that the graph metrics retained information about the encoding states even after controlling for window-wise FD. However, the multivariate analysis using the subnetwork-wise local efficiency adjusted for the window-wise FD did not provide significant classification accuracy (70%, p=0.1625, permutation test).

Second, we repeated the analysis with including only half of the participants (n = 13) such that the difference in mean FD between the high and low encoding states became minimal in the subsample. In this subset of participants, mean FD was closely matched between the two states (high: 0.1265 ± 0.0142, low: 0.1265 ± 0.0129; signed rank = 39, p=0.6848), and the window-wise FD was not correlated with either memory encoding performance (Pearson's $r$ = 0.0032 ± 0.1077; signed rank = 47, p=0.9460) or global efficiency (Pearson's $r$ = −0.0778 ± 0.2716; signed rank = 33, p=0.4143). The difference in global efficiency between the two states remained significant (signed rank = 89, p=0.0007; *Supplementary file 1W*), indicating that the findings with this subset could not be explained by motion-derived confounds. However, the difference in modularity between the states was not significant in this subsample (signed rank = 65, p=0.1909), possibly implying that the relationship between modularity and memory encoding performance is less robust compared with that of global efficiency.

Taken together, the results of these analyses indicate that our finding of large-scale integration associated with memory encoding performance cannot be accounted for by motion confounds. On the other hand, although the modularity difference between the states is statistically significant in our main analysis, that is less robust against one of the above supplement analyses compared with the network integration.

# Discussion

We demonstrated dynamic reconfiguration of a large-scale functional brain network associated with temporal fluctuations in encoding performance. Importantly, we observed a higher level of network integration during periods of high (vs. low) encoding performance. This effect was mainly driven by increased inter-subnetwork integration of the subcortical, default mode, salience, and visual networks with other subnetworks. Furthermore, dynamic reconfiguration of functional brain network architecture was uniquely related to encoding performance, and not accounted for by the effect of simple visual stimulation or RT for semantic judgment.

## Time-varying FC among memory-related brain regions

Previous neuroimaging studies have repeatedly shown that successful memory encoding is associated with activation/deactivation of specific brain regions, particularly the SME regions (including the medial temporal lobe and prefrontal cortex) (*Wagner et al., 1998*; *Brewer et al., 1998*; *Paller and Wagner, 2002*; *Reber et al., 2002*; *Uncapher and Rugg, 2005*; *Kim, 2011*) and SFE regions (including the posterior cingulate cortex and temporoparietal junction) (*Wagner and*

*Davachi, 2001*; *Otten and Rugg, 2001*; *Daselaar et al., 2004*; *Kim, 2011*). However, the ways in which the dynamic interaction among these regions supports successful memory encoding remain unclear. By analyzing time-varying FC within 36 s time windows, we showed that FC between the hippocampus and occipital cortex, both regarded as SME regions, was higher during periods of high (vs. low) encoding performance. This may indicate that successful encoding of visual information is supported by functional interactions between the visual area and the hippocampus, a key structure for memory formation (*Eichenbaum et al., 2007*).

Although we hypothesized to observe a general increase in within-subnetwork connectivity for the SME, this was not the case. FC increase was observed specifically between the hippocampus and occipital cortex, and not among all nodes of the SME regions. One possible explanation for this finding is that our SME ROIs were defined on the basis of a meta-analysis of task activation, not FC. It is possible that the coordinates identified by local task-activation analysis (i.e., the SME/SFE) are not optimal for analyzing FC patterns related to memory encoding performance.

## Dynamic reconfiguration of a large-scale functional brain network

Successful memory encoding is likely to be influenced by many state factors, such as arousal, attention to external stimuli, and motivation to perform a task (*Chun and Turk-Browne, 2007*; *Gruber et al., 2014*; *Tambini et al., 2017*). Therefore, temporal fluctuations in encoding performance may be associated with dynamic interactions among a diverse set of brain systems, beyond the so-called memory system. In the current study, we observed differential FC patterns between the high and low encoding states, both within and across many subnetworks. Importantly, during the periods of high (vs. low) encoding performance, FC was increased between specific brain regions, whereas it was decreased between another specific set of regions (see *Supplementary file 1D and 1E* for complete lists). In addition, we observed a marked increase in FC between distant brain regions, whereas FC decreases were prominent between (but not limited to) neighboring regions. These findings suggest a systematic reconfiguration of the large-scale functional brain network related to incidental encoding performance, rather than a uniform increase/decrease in FC across the entire network. Our results indicate that enhanced long-range interregional FC is important for better encoding performance (which was also confirmed by graph analysis). At the same time, the finding that the connections showing decreased FC during the high encoding state outnumbered those showing increased FC (687 vs. 98) highlights the importance of selective functional decoupling. One interesting possibility is that 'breaking' particular FC patterns, such as those observed during task-free resting states, may play an active role in successful memory encoding. This possibility could be tested in future studies by directly comparing dynamic FC patterns during memory encoding task and those during the resting state (*Wang et al., 2016*).

Recent studies have shown the dynamic nature of FC patterns in the brain and their contributions to a variety of cognitive functions (*Bassett et al., 2011*; *Cole et al., 2014*; *Bassett et al., 2015*; *Sadaghiani et al., 2015*; *Wang et al., 2016*; *Mohr et al., 2016*; *Cohen, 2017*; *Kucyi et al., 2018*). For example, one study reported that dynamic FC in a time window of 40 s reflected temporal fluctuations in arousal level as indicated by RTs in a continuous performance task, and spontaneous eyelid closure (*Wang et al., 2016*). Other studies have also shown that dynamic reconfigurations of FC patterns are observed across many situations, from performance of cognitively demanding tasks (e. g., working memory and Stroop tasks) to simple perceptual detection of visual and auditory stimuli (*Gonzalez-Castillo et al., 2015*; *Cohen and D'Esposito, 2016*; *Spielberg et al., 2015*; *Sadaghiani et al., 2015*; *Godwin et al., 2015*). Our findings extend previous studies by showing that temporal fluctuations in FC across a large-scale brain network are related to incidental memory encoding.

Although our finding of large-scale integration associated with memory encoding is novel, several previous studies investigated FC changes during memory encoding. For example, a previous study reported increased FC between the hippocampus and neocortical regions including the occipital cortex during successful (vs. unsuccessful) memory encoding (*Ranganath et al., 2005*). It should be noted that FC between the hippocampus and cortical areas is also considered to be important for memory retrieval (*King et al., 2015*; *Ritchey et al., 2013*). *Rugg and Vilberg, 2013* suggested that the MTL and several cortical areas such as the posterior cingulate, ventral parietal, and medial prefrontal cortices constitute a 'general recollection network,' which plays a key role in successful memory retrieval across various contexts (*Rugg and Vilberg, 2013*). One recent study examining FC

patterns during memory retrieval reported greater FC among many brain regions during correct (vs. incorrect) retrieval (*Schedlbauer et al., 2015*). This finding may suggest that successful memory encoding and retrieval are at least partially supported by common patterns of network dynamics. However, it has also been reported that functional coupling between the hippocampus and specific regions (e.g., the posterior cingulate, ventral parietal, and medial prefrontal cortices) differs between memory encoding and retrieval (*Huijbers et al., 2011*). Future studies should directly compare large-scale network configurations during encoding and retrieval in a single experiment.

## Network integration and segregation associated with incidental encoding performance

Using graph analysis, we tested whether dynamic changes in network integration/segregation are associated with encoding performance. At the entire-network level, we found a higher degree of integration (as measured by global efficiency) during periods of high encoding performance. When we examined individual subnetworks, we observed greater integration during the high encoding state in the SAN and SUB subnetworks. Importantly, the SAN was the only subnetwork showing a significant state (high vs. low)-by-metric (integration vs. segregation) interaction, indicating that subnetwork-wise integration (but not segregation) in the SAN is related to better memory encoding performance. In contrast, we did not observe a state-by-metric interaction in the SUB, likely because the subnetwork-wise local efficiency also showed a trend indicating an increase. The importance of the SUB in network integration was supported by the results of another measure of inter-subnetwork integration ($E_{is}$), indicating that the SUB was the only subnetwork exhibiting significantly higher $E_{is}$ during the high encoding state with all other subnetworks. Importantly, $E_{is}$ and PCs characterize different aspects of a network: the former quantifies integration between a specific pair of subnetworks, whereas the latter quantifies the diversity of inter-subnetwork connections of a particular subnetwork/node to all other subnetworks (*Rubinov and Sporns, 2010*; *Power et al., 2013*; *Marek et al., 2015*; *Shine et al., 2016*). In the current case, the results of these two metrics convergently suggested that integration of the SUB with other subnetworks is associated with successful memory encoding. Therefore, the subcortical nodes (i.e., the thalamus and putamen) may play a unique role in incidental memory encoding, contributing to both within- and across-subnetwork functional interactions and serving as a hub to support large-scale network integration (*Bell and Shine, 2016*).

Following the SUB, the DMN was the second notable subnetwork showing higher inter-subnetwork integration with many subnetworks (i.e., the CON, VIN, FPN, SAN, and SUB) during the high encoding state. Previous studies have shown the involvement of the DMN in episodic memory (*Greicius and Menon, 2004*; *Spreng and Grady, 2010*; *Andrews-Hanna et al., 2010*). It has also been reported that FC within the DMN is associated with subsequent memory performance (*Simony et al., 2016*; *Hasson et al., 2009*). Consistent with these previous studies, we observed significant FC increases within the DMN, including among the superior frontal gyrus and angular gyrus. In addition, in one of the DMN submodules located in the mPFC identified by group-level modular decomposition (*Fornito et al., 2012*), we found a trend indicating increased local efficiency during periods of better encoding performance. This result may imply that increased FC within a specific submodule of the DMN is related to memory encoding, although further research is needed to elucidate possible differentiation among the DMN submodules. Importantly, the current study extends previous findings by revealing that inter-subnetwork integration between the DMN and other subnetworks was related to within-individual time-to-time fluctuations of memory performance, providing further evidence of a central role of the DMN in successful memory encoding. Notably, some regions within the DMN (e.g., the posterior cingulate cortex and temporoparietal junction) exhibited the SFE (i.e., decreased trial-related activation for successful [vs. unsuccessful] memory encoding). This highlights the importance of examining both trial-related activation changes and FC changes to understand the role of specific brain systems in certain cognitive functions.

Although our analysis primarily targeted dynamic FC patterns in the timescale of 30–40 s, we observed similar results in other window sizes (i.e., 7.2–60 s). This suggests a reliable association between large-scale network integration and memory encoding performance across a range of timescales. However, it should be noted that even the shortest time windows in our analysis (i.e., 7.2 s) included multiple picture trials. Therefore, our findings are related to the link between particular states of large-scale networks and average memory performance within given time periods, rather

than the neural processes underlying single memory-encoding events. To clarify the role of large-scale network states in individual memory encoding events, future research could employ trial-by-trial analysis of FC patterns. For instance, a previous study investigated trial-by-trial FC patterns during a 6 s (4 TRs) period before auditory stimulus presentation, reporting that the pre-stimulus FC patterns differentiated upcoming perceptual detection performance (*Sadaghiani et al., 2015*). This type of analysis could also be useful for understanding large-scale network states predictive of single-item subsequent memory. Furthermore, given the rapid nature of memory-encoding processes (*Viskontas et al., 2006*), large-scale network states with much shorter timescales (e.g., less than a second) may also be related to individual memory-encoding events. Recent human electrophysiological studies have shown that large-scale neural synchronization with timescales of hundreds of milliseconds across many regions (e.g., MTL, ventral parietal, and prefrontal cortices) are associated with both successful memory encoding and retrieval (*Watrous et al., 2013*; *Solomon et al., 2017*). Bridging observations across multiple timescales, from milliseconds to minutes, represents an important next step in research of large-scale network dynamics in episodic memory.

## Multivariate pattern classification using graph metrics

Our multivariate analysis using graph metrics demonstrated that functional network architecture during the high and low encoding states can be reliably classified using subnetwork-wise metrics of integration. That is, the graph metrics defined at the subnetwork level contain sufficient information to distinguish the high from low encoding states. When we used the entire connectivity patterns as the input, the classification accuracy dropped to chance levels. This suggests that the use of graph metrics can efficiently reduce the number of features and achieve more accurate predictions. The method employed here could be useful for many other applications, such as comparing large-scale brain networks between specific disease groups and normal controls. In addition to other methods of connectivity pattern classification (*Sadaghiani et al., 2015*; *Rosenberg et al., 2016*; *Hein et al., 2016*), multivariate analysis using graph metrics could facilitate future research on large-scale brain network architecture.

## Methodological considerations for graph analysis

Several details of the analysis should be noted. First, recent studies have suggested that motion confounds may affect temporal fluctuations in FC patterns (*Laumann et al., 2017*). To address this concern, we performed two supplement analyses, both of which supported our main findings regarding large-scale integration associated with memory encoding performance. Although it was not significant, we observed a trend indicating a negative correlation between FD and memory performance. One possible explanation for this negative correlation is that some individuals may have exhibited a fluctuating level of arousal over the course of the task, which could be positively correlated with window-wise memory performance and negatively correlated with mean FD (*Laumann et al., 2017*). Such momentary fluctuations in arousal level would be inevitable for some individuals, particularly during tasks that require continuous performance (like the one we used). The fact that we observed a significant difference in global efficiency between the high and low encoding states even after excluding such participants provides strong support for our finding. Although motion confounds would be expected to influence both momentary encoding performance and dynamic FC patterns, our supplement analyses suggest that the finding of large-scale integration associated with encoding performance was above and beyond the effect of motion confounds.

Second, the present results concerning higher modularity during the high encoding state should also be considered. Although we observed increased global efficiency during the high encoding state, this was not accompanied by decreased modularity. Unlike global efficiency, modularity is a metric based on the community structure of a network (*Rubinov and Sporns, 2010*; *Wig, 2017*). Thus, the current results may suggest that the number of long-range connections across modules was increased, rather than indicating that the network became less modular. Importantly, theoretical studies have shown that a network can exhibit both high integration and high modularity simultaneously through sparse long-range connections across communities (*Tononi et al., 1994*; *Watts and Strogatz, 1998*; *Pan and Sinha, 2009*; *Meunier et al., 2010*). Recent empirical studies of brain networks have also shown that integration and segregation are not necessarily incompatible with each other (*Mohr et al., 2016*; *Bertolero et al., 2018*). In accord with this perspective, we found

simultaneous increases in the proportion of edges within the SUB and those across the SUB and the other subnetworks during the high encoding state. This suggests that the SUB may have contributed to entire network-level segregation (i.e., modularity) by its increased within-subnetwork connections, and also contributed to inter-subnetwork integration by its increased long-range connection with other subnetworks. However, the results regarding the modularity difference associated with memory encoding performance appeared to be less robust compared with those of global efficiency (i.e., the analysis based on a subset of participants whose FD was less dependent on memory performance). Further methodological improvements are needed to effectively identify possible coexistence of integration and segregation in large-scale brain networks related to various cognitive functions including memory encoding (*Lord et al., 2017*).

Third, our additional analysis controlling for overall FC strength suggested that window-to-window fluctuations in the graph metrics substantially covaried with overall FC strength. This makes it difficult to disentangle the effects of network integration/segregation from those of overall FC strength in examining dynamic changes in functional brain networks. However, this does not necessarily undermine the usefulness of graph analysis, for the following reasons. First, correcting for overall FC strength would be overly strict: although it rules out the possibility that observed differences in graph metrics result from artifacts of spurious weak connections, it could also remove real differences in network architecture (*van den Heuvel et al., 2017*). Second, even if graph metrics and overall FC strength are highly correlated, subnetwork-wise graph metrics may provide additional insight into detailed network organization (e.g., specificity and heterogeneity among subnetworks), which may not be captured by overall FC strength. Importantly, we examined the reliability of edges across time windows, and confirmed that our findings are not attributable to differential proportions of false-positive edges (*Zalesky et al., 2014*; *van den Heuvel et al., 2017*). In addition, the dynamic reconfiguration of the large-scale network associated with encoding performance was also supported by an analysis that did not rely on graph metrics. Overall, it is unlikely that our findings were solely due to temporal fluctuations in overall FC strength.

## Limitations and future perspectives

Although our study provides a number of novel findings about the dynamic FC associated with incidental memory encoding, several limitations should be considered. First, the current study was unable to determine the causal directions between dynamic FC and temporal fluctuations in encoding performance. Recent studies have shown that large-scale FC patterns with timescales of 30–40 s show dynamic fluctuations even in the resting state (*Calhoun et al., 2014*; *Allen et al., 2014*; *Zalesky et al., 2014*; *Betzel et al., 2016*; *Shine et al., 2016*). Given these previous findings, it could be hypothesized that intrinsic, spontaneous dynamics of FC patterns underlie temporal fluctuations in encoding performance. However, we cannot rule out the possibility that different levels of encoding performance across the time windows induce time-varying FC patterns. Future research should be conducted to test these two possibilities using methods that can causally manipulate large-scale FC patterns (*Ezzyat et al., 2017*; *Muldoon et al., 2016*).

Second, to examine dynamic changes in FC patterns, we divided the fMRI time series into time windows, and sorted the windows in reference to behavioral data (i.e., encoding performance). As a result, we sorted the FC patterns into two 'states.' However, this does not necessarily mean that there are only two dynamic FC states; it is possible that there are more than two dissociable states and only some of them are truly related to encoding performance. Some recent studies have employed other approaches, first identifying distinct dynamic FC states based solely on neural data, then relating individual states to behavioral measures (*Calhoun et al., 2014*; *Shine et al., 2016*; *Wang et al., 2016*). Such approaches could provide further detail about the relationships between dynamic network architecture and memory encoding. Meanwhile, in the current study, comparison of the FC patterns of high and low encoding states revealed very similar patterns. This implies that approaches attempting to identify distinct dynamic states solely using neural data may not work for our data. However, approaches using behavioral data as references for classification may be particularly useful when behavioral performance is associated with very subtle differences in network states, as in the current study.

Third, although our study demonstrated large-scale integration associated with temporal memory performance within individuals, we did not examine how the network characteristics were related to inter-individual variations in memory performance. To further clarify the role of large-scale networks

in memory encoding, future studies should investigate how these networks are related to individual differences in memory performance in both healthy and clinical populations.

To summarize, we analyzed time-varying FC patterns during an incidental encoding task, and found that dynamic reconfiguration of a large-scale functional brain network was associated with encoding performance. The periods of high encoding performance were characterized by greater network integration, mainly driven by inter-subnetwork integration between the subcortical, default mode, salience, and visual networks. Our findings provide a better understanding of the neural mechanisms of memory encoding, highlighting the importance of orchestration across many distinct brain systems.

## Materials and methods

### Participants

A total of 30 university students (20 males; age 18–22 years, mean ± SD = 20.0±1.2) participated in the study after providing written informed consent. Four participants who fell asleep in the scanner and did not respond in more than 20 trials were excluded from the analysis. One additional participant who did not follow the instructions (not making any 'low confidence' responses in the surprise memory test) was also excluded. The remaining 25 participants (17 males; age 18–22 years, mean ± SD = 20.1±1.1) were therefore available for the analysis. All experimental procedures were approved by the Ethics Committee of Kochi University of Technology.

### Stimuli

Stimuli consisted of color pictures (sized 8° × 6°) and a white fixation cross (sized 0.8° × 0.8°). The pictorial stimuli included 360 pictures showing man-made objects (e.g. commodities, stationery, musical instruments, and appliances) and 360 pictures showing natural objects (e.g. animals, plants, fruits, and natural scenes). These pictures were selected from the Bank of Standardized Stimuli (BOSS) (*Brodeur et al., 2010*) and a commercially available image database. All color pictures underwent luminance, contrast, and spatial frequency equalizing by in-house MATLAB (MathWorks, Natick, MA) code (freely available at https://github.com/Ruedeerat/RGBshine/, *Keerativittayayut, 2018*; copy archived at https://github.com/elifesciences-publications/RGBshine) adapted from the SHINE toolbox (*Willenbockel et al., 2010*). Half of the pictures (180 man-made and 180 natural pictures) were randomly selected for use in the incidental memory encoding task. The remaining 360 pictures were used as unstudied pictures in the surprise memory test. The tasks were programmed and administered using Presentation software (Neurobehavioral Systems, Berkeley, CA). We projected the visual stimuli on a screen located behind the scanner. Participants viewed the projected visual stimuli through a mirror attached to a head coil.

### Experimental paradigm

The experimental paradigm (*Figure 1A*) was based on the subsequent memory approach, which has been widely used in previous research (*Wagner et al., 1998*; *Paller et al., 1987*). Participants took part in a two-stage experiment: an incidental memory encoding task followed by a surprise memory test. During the incidental memory encoding scans, participants studied the pictorial stimuli. Twenty minutes later, memory for the studied pictures was assessed by the surprise memory test outside the scanner.

In the incidental memory encoding task, participants studied 360 pictures in three runs. Each run began with a central fixation cross for 15 s, followed by a continuous series of 180 rapidly intermixed trials. Sixty man-made picture trials, 60 natural-made picture trials, and 60 fixation trials were pseudo-randomly presented with counterbalancing (each trial type followed every other trial type equally often). Each run ended with an additional fixation period of 20 s. For a picture trial, a pictorial stimulus was presented on the screen for 2500 ms, followed by a 500 ms presentation of a fixation cross. For a fixation trial, only a fixation cross was presented for 3 s. During the picture trials, participants were instructed to make a semantic judgment (man-made or natural) by right-handed button press as soon as possible after the picture onset. The total time for performing the incidental encoding task was approximately 30 min.

In the surprise memory test, participants were presented with the 360 studied pictures from the incidental memory encoding task, as well as 360 unstudied pictures. They were asked to indicate whether they recognized each picture as studied with high confidence, studied with low confidence, or unstudied. Each picture was displayed individually with self-paced timing. Participants responded by right-handed keyboard press.

## Image acquisition and preprocessing

All scanning procedures were performed using a 3T Siemens Verio MRI scanner (Siemens, Erlangen, Germany) equipped with a 32-channel head coil. A high-resolution T1-weighted anatomical image was collected for each participant (MPRAGE; repetition time [TR]=2500 ms; echo time [TE]=4.32 ms; flip angle [FA]=8°; field of view [FOV]=230 mm; matrix = 256 × 256; in-plane resolution = 0.9 × 0.9 mm$^2$; slice thickness = 1 mm; 192 slices; acceleration factor = 2). Functional data were collected using a multiband echo planar imaging (EPI) pulse sequence (TR = 720 ms; TE = 33 ms; FA = 52°; FOV = 192 mm; matrix = 64 × 64; in-plane resolution = 3 × 3 mm$^2$; slice thickness = 3 mm; slice gap = 0.75 mm; 45 slices; multi-band acceleration factor = 5), which afforded whole-brain coverage (*Xu et al., 2013*). Preprocessing was carried out using SPM12 (Wellcome Department of Cognitive Neurology, London, UK). The first five volumes of each run were discarded before preprocessing. The remaining functional volumes were spatially realigned, coregistered to the individual high-resolution anatomical image, normalized to Montreal Neurological Institute (MNI) space, spatially smoothed with 8 mm full width at half maximum (FWHM) Gaussian kernel, and resampled to a spatial resolution of 2 × 2 × 2 mm$^3$.

## Regions of interest

In the current study, we used two different sets of ROIs. The first set of ROIs was used to investigate FC patterns among well-established memory-related brain regions. Therefore, we used a set of 21 ROIs derived from a recent meta-analysis of the SME/SFE (*Kim, 2011*). The ROIs included 11 brain regions associated with the SME (e.g., the inferior frontal cortex, hippocampus, intraparietal sulcus, and middle occipital gyrus) and 10 brain regions associated with the SFE (e.g., the frontal pole, superior temporal gyrus, posterior cingulate cortex, and temporoparietal junction; see *Supplementary file 1A* for the list of all 21 ROIs). The second set of ROIs was used to investigate FC patterns across a large-scale brain network. We used 224 ROIs consisting of 10 subnetworks from the Power atlas (*Power et al., 2011*). The subnetworks had the following labels: sensorimotor networks (SMN), cingulo-opercular network (CON), auditory network (AUD), default mode network (DMN), visual network (VIN), fronto-parietal network (FPN), salience network (SAN), subcortical nodes (SUB), ventral attention network (VAN), and dorsal attention network (DAN) (see *supplementary file 1C* for the list of the ROIs). Although the Power atlas was originally derived from resting-state fMRI data, the same set of ROIs and subnetwork labels have been repeatedly used in task-fMRI studies on large-scale functional brain networks (*Cole et al., 2014*; *Cohen et al., 2014*; *Sadaghiani et al., 2015*; *Cohen and D'Esposito, 2016*; *Mohr et al., 2016*; *Westphal et al., 2017*). To cross-validate our findings regarding large-scale networks, we also used 285 ROIs organized into 11 subnetworks derived from the Gordon atlas (*Gordon et al., 2016*; *Supplementary file 1R*).

## Trial-related activation analysis

To identify brain regions showing the SME (i.e., greater activation in HH than Miss trials) and the SFE (i.e., greater activation in Miss than HH trials), we performed trial-related activation/deactivation analysis using a general linear model (GLM). First, based on participants' responses in the surprise memory test, we categorized the 360 picture trials of the incidental encoding task into three types: high-confidence hit (HH, subsequently remembered with high confidence), low-confidence hit (LH, remembered with low confidence), and Miss (forgotten) trials. Second, we constructed a GLM that included trial-related regressors denoting: (1) the onsets of HH trials, (2) the onsets of LH trials, and (3) the onsets of Miss trials, following the conventions of the subsequent memory approach (*Wagner et al., 1998*). Each trial was modeled using a box-car function (initiating at picture onset, duration = 2500 ms) convolved with a canonical hemodynamic function provided by SPM12. The GLM also included eight nuisance regressors per run: six motion parameters as well as mean time series in the white matter (WM) and cerebral spinal fluid (CSF). The mean time series in the WM and

CSF were obtained by averaging time series of voxels within the WM and CSF masks, each of which was derived from an individual's segmented structural image (binarized at a threshold of tissue probability >0.8) (*Biswal et al., 2010*; *Vahdat et al., 2011*). The second-level random-effects analysis (one-sample t-tests) was performed using contrast images derived from individual participants (i.e., HH minus Miss for the SME and Miss minus HH for the SFE). The statistical threshold was set at voxel-wise p<0.05, family-wise error corrected across voxels with the gray matter (defined by 'TPM. nii' implemented in SPM12, thresholded at 0.5). For a set of selected ROIs (*Figure 3B*), we extracted beta estimates of individual participants and contrasts from 5 mm radius spheres centered on the MNI coordinates derived from the meta-analysis of the SME/SFE (*Kim, 2011*).

## Extraction of fMRI time series

We extracted residual time series data from each ROI using a voxel-wise GLM, in accord with previous research (*Cao et al., 2014*; *Tompary et al., 2015*; *Shine et al., 2016*; *Cohen and D'Esposito, 2016*). More specifically, we averaged time series across voxels within a 5 mm radius sphere around each ROI, after regressing out the trial-related (HH, LH, and Miss) and nuisance (six motion parameters as well as WM and CSF, as well as their temporal derivatives and quadratic terms) signals defined by the regressors of the aforementioned GLM (*Power et al., 2014*; *Ciric et al., 2017*). Note that we did not include global signal regression because it could introduce spurious anti-correlations. The obtained residual time series were used for FC analysis described below. For scrubbing, frames with FD >0.2 mm were censored (*Power et al., 2012*; *Power et al., 2014*; *Laumann et al., 2017*), and ignored when computing FC. For the additional control analysis using trial-evoked time series (*Supplementary file 1K*), we regressed out only the nuisance signals (i.e., motion parameters and WM/CSF time series), while maintaining the trial-related signals. All other procedures were identical to the main analysis.

## Definition of time windows

We sought to examine time-varying FC patterns associated with incidental memory encoding performance. To do so, we first divided the extracted time series into 36 s (i.e., 50 TRs) time windows, resulting in 45 windows per participant. This window size was determined on the basis of recent studies showing dynamic changes in FC in time periods of 30–40 s (*Braun et al., 2015*; *Sadaghiani et al., 2015*; *Wang et al., 2016*; *Mohr et al., 2016*). Importantly, we confirmed that our findings were robust to a range of window sizes (7.2–60 s; see *Supplementary file 1H*). We also confirmed that our results were unchanged when we used overlapping sliding windows (sliding in steps of 1 TR, resulting in 2100 windows per participant) or when we used time windows shifted by 5 s (with taking into account the hemodynamic delay; see *Supplementary file 1I and 1J*). Next, for each participant, we classified the time windows into either 'high encoding' or 'low encoding' states based on window-wise encoding performance: the proportion of HH trials (the number of HH trials divided by that of picture trials) computed within each window. We used participant-specific median values for the classification, ensuring roughly equal numbers of windows for the high and low encoding states at an individual level. When a window had the exactly the same value as the median, we classified the window into either the high or low encoding state, depending on each participant, so that we could maximally equate the number of windows between the two states. In additional analyses, we also used tertiles and quartiles (instead of medians) to classify the windows according to memory encoding performance (*Supplementary file 1M and 1N*). To examine history dependence in the encoding states, we computed the probability of state switching (i.e., high to low or low to high, as opposed to high to high or low to low). We used a permutation test to determine statistical significance: the (group-averaged) probabilities of state switching for the empirical data were compared with a null distribution derived from 1000 permutations (i.e., we permuted the order of 45 windows within each participant and computed the probability of state switching for the permuted sequences of windows).

To test the robustness and specificity of our findings, we repeated the classification analysis with several alternative inputs. First, we classified the windows based on the proportion of HH and LH trials (i.e., the number of HH plus LH trials divided by that of the picture trials). This analysis confirmed that our findings held true when we included the LH trials in computing the window-wise encoding performance (see *Supplementary file 1L*). Second, to rule out the possibility that our findings

resulted from simple visual-related brain responses, we classified the windows into 'more pic' and 'fewer pic' periods, based on the proportion of the picture trials (i.e., the number of picture trials divided by the total number of trials including fixation trials) irrespective of encoding performance. Third, to assess the influence of window-to-window variability in RT for semantic judgment (possibly reflecting task difficulty or general arousal level not directly related to memory performance), we classified the windows into 'longer RT' and 'shorter RT' periods based on mean RT computed within each window. The RT of a trial was defined as the time from the picture onset to the participant's button press (1167.1 ± 240.8 ms, mean ± SD across participants). The results from the second and third analyses confirmed that our findings are specific to encoding performance (see *Supplementary file 1P and 1Q*).

## FC analysis

We examined how FC patterns among ROIs (either in the 21-node or 224-node networks) differed between the high and $_{low}$ encoding states. For each time window, we computed Pearson's correlation coefficients of the time series between all pairs of ROIs, which were Fisher Z-transformed to form a connectivity matrix. We then averaged the connectivity matrices across the windows, separately for the high and low encoding states. For statistical tests of the difference in FC patterns between the states, we used Wilcoxon's signed-rank tests across participants. The significance threshold was set at p=0.05, with multiple comparison corrections controlling for FDR. To compute Euclidean distance, we used *x*, *y*, and *z* of MNI coordinates for each ROI. We used BrainNet Viewer (*Xia et al., 2013*; RRID:SCR_009446) to visualize FC changes.

## Graph analysis

We performed graph analysis to examine integration and segregation of the 224-node network, using the Brain Connectivity Toolbox (*Rubinov and Sporns, 2010*; RRID: SCR_004841). Note that this analysis was not applied to the 21-node network because graph metrics estimated in small networks are not necessarily stable (*Rubinov and Sporns, 2010*; *Sadaghiani et al., 2015*). To derive graph metrics from the 224-node network, we constructed an unweighted, undirected graph from a $224 \times 224$ connectivity matrix by applying a proportional threshold of connection density = 0.15. To ensure that effects were not driven by the particular connection density, we checked robustness by varying the threshold values: 0.1, 0.15, 0.2, and 0.25 (*Supplementary file 1H*).

Network topologies were characterized using the following metrics: global efficiency ($E_g$), local efficiency ($E_{loc}$), inter-subnetwork efficiency ($E_{is}$), and PC. In the present study, $N$ is the set of all nodes in the network, and $n$ is the number of nodes. $(i, j)$ is a link between nodes $i$ and $j$, $(i, j \in N)$. $a_{ij}$ is the connection status between $i$ and $j$: $a_{ij} = 1$ when link $(i, j)$ exists; $a_{ij} = 0$ when no connection is present. $d_{ij}$ is the shortest path length between nodes $i$ and $j$. $M$ is the set of subnetworks, and $m$ is the number of subnetworks.

The global efficiency ($E_g$) is a measure of integration. A network with high $E_g$ is considered topologically integrated. The global efficiency of a network is the average of the inverse shortest path lengths across all pairs of nodes:

$$E_g = \frac{1}{n} \sum_{i \in N} \frac{\sum_{j \in N, j \neq i} d_{ij}^{-1}}{n-1}$$

The local efficiency ($E_{loc}$) is a measure of segregation. The local efficiency of node $i$ is the average of the inverse shortest path lengths defined in the subgraph consisting of $i$ and its neighboring nodes

$$E_{loc} = \frac{\sum_{j,h \in N, j \neq i} a_{ij} a_{ih} \left[ d_{jh}(N_i) \right]^{-1}}{k_i(k_i - 1)}$$

where $k_i$ is the number of links connected to $i$, and $d_{jh}(N_i)$ is the shortest path length between $j$ and $h$, that contains only neighbors of $i$. For a network- or subnetwork-level measure of segregation, $E_{loc}$ is averaged across nodes within a network or subnetwork, respectively.

The participation coefficient ($PC$) is an alternative measure of integration, which quantifies the diversity of inter-subnetwork connections of a node:

$$PC = 1 - \sum_{m \in M} \left( \frac{k_i(m)}{k_i} \right)^2$$

where $k_i(m)$ is the number of links between $i$ and all nodes in subnetwork $m$. For a subnetwork-level measure, $PC$ is averaged across nodes within a subnetwork.

Furthermore, we defined 'inter-subnetwork' efficiency ($E_{is}$) as a measure of integration between a specific pair of subnetworks:

$$E_{is} = \frac{1}{s} \sum_{i \in S} \frac{\sum_{j \in T} d_{ij}^{-1}}{t}$$

where $S$ and $T$ are the (non-overlapping) sets of nodes in two subnetworks, and $s$ and $t$ are the numbers of nodes in them. Note that $d_{ij}$ is defined over the entire network, and the shortest path may be mediated by nodes outside the subnetworks of interest.

We also computed modularity as an index of how well a network can be partitioned into distinct communities:

$$Q = \sum_{u \in M} \left[ e_{uu} - \left( \sum_{v \in M} e_{uv} \right)^2 \right]$$

where the network is partitioned into a set of non-overlapping modules $M$ (identified by Louvain's algorithm), and $e_{uv}$ is the proportion of all edges that connect nodes in module $u$ with nodes in module $v$ (**Rubinov and Sporns, 2010**; **Fornito et al., 2012**). Although the Louvain algorithm is stochastic, we confirmed that our results were stable over iterations (mean $\pm$ SD of $z$ value [Wilcoxon signed-rank test] over 5,000 iterations = 2.6051 $\pm$ 0.0709 for the modularity difference between the high and low encoding states). In addition, modularity computed with the Louvain algorithm was strongly correlated with that computed by the Newman (deterministic) algorithm from window to window (Pearson's $r$ = 0.9796 $\pm$ 0.0112, $z_{(24)}$ = 4.3724, $P$ = 1.2290 $\times$ 10$^{-5}$). This confirmed that our choice of algorithm did not affect our findings.

The modularity contribution of each subnetwork was calculated as $e_{uu} - \left( \sum_{v \in M} e_{uv} \right)^2$ (**Sadaghiani et al., 2015**). Because this analysis requires a fixed community assignment of nodes across all windows and participants, we used the pre-defined subnetwork labels of the Power atlas (unlike all other analyses involving modular structure, for which we used the Louvain algorithm for optimal community detection).

These graph metrics were calculated for each window, then averaged across the windows, separately for the high and low encoding states. Finally, the graph metrics were compared between the two states across 25 participants using Wilcoxon signed-rank test. All statistical results were corrected for multiple comparisons using FDR correction.

We investigated whether the nodes belonging to the DMN could be further divided into submodules during the high and low encoding states. Using a modular decomposition technique (**Fornito et al., 2012**), we first computed functional connectivity across the 56 DMN nodes and construct an unweighted, undirected graph from a 56 $\times$ 56 correlation matrix, separately for each time window and participant. For each window, we applied the Louvain method to decompose the DMN nodes into submodules. Community affiliation indices, the outputs from modularity decomposition, were submitted to construct a 56 $\times$ 56 coclassification matrix by defining $C_{ij}$ = 1 if node $i$ and $j$ belonged to the same submodule and $C_{ij}$ = 0 otherwise. We then generated group consistency matrices (G) by counting how frequently each node was assigned to be the same submodule across all windows, separately for groups of high and low encoding windows. Finally, the consistency matrices (G) were submitted to a group level modular decomposition to classify nodes which were more likely to belong to the same module across participants.

To rule out the possible confounding effects of time, within and across sessions, we performed additional statistical analyses, as follows (**Supplementary file 1O**). First, to exclude the effects of the amount of time passed within each session, we define a 45-by-1 dummy vector denoting the order of windows within each session (i.e., [1, 2, 3, . . . 15], repeated three times), and regressed out this effect on a window-by-window basis before averaging graph metrics within each state. Second, to

exclude the effect of the amount of time passed across sessions, we defined another 45-by-1 vector denoting session (i.e., [1, 1, … 2, 2, … 3, 3, …]), and regressed out this effect from graph metrics of each window. We confirmed that neither linear nor quadratic effects of the amount of time passed explained our results.

It should be noted that a recent paper raised a concern about the possible influences of overall FC strength on graph metrics (*van den Heuvel et al., 2017*). In short, the authors argued that weaker overall FC of a network may result in the inclusion of more random connections (particularly when a graph is constructed using proportional thresholding), which tends to give a higher value of global efficiency and a lower value of local efficiency. In other words, differences in graph-metric values between two networks may reflect differences in overall FC strength. To address this concern, we performed an additional analysis controlling for the effect of overall FC strength, as proposed in the paper (*van den Heuvel et al., 2017*). Specifically, we first computed overall FC strength (the mean of all positive values across all elements of a connectivity matrix) for each window. We then regressed out the overall FC strength from all graph metrics to obtain 'adjusted' graph metrics. We performed the statistical analysis using these adjusted graph metrics in the same manner as the main analysis (*Supplementary file 1V*).

We also analyzed 'edge reliability' to confirm that the difference in the proportions of reliable edges between the high and low encoding states did not affect our results. Specifically, for each participant and state, we examined how often an edge appeared between a given node pair across time windows. An edge was defined as 'reliable' if it consistently appeared across windows more than by chance. To determine the chance level, we created 100 randomized networks from each of the real networks per participant and state, while preserving degree distributions (*Rubinov and Sporns, 2010*), and generated null distributions of the probability of edge appearance. The 95th percentile of this null distribution was used as a threshold to determine the reliable edges in the real networks. We compared the proportions of reliable edges (i.e., the number of reliable edges relative to all possible edges) between the high and low encoding states using a signed-rank test across participants.

### Multivariate pattern analysis

For the multivariate analysis based on graph metrics, we performed across-participant binary classification (with leave-one-out cross validation) using a support vector machine (SVM) implemented in LIBSVM (*Chang and Lin, 2011*; RRID:SCR_010243). We used PCs and local efficiency of the 10 subnetworks and inter-subnetwork efficiency of 45 subnetwork pairs (averaged across windows for each participant and state) as inputs for the classifier. The input variables were Z-score normalized (mean = 0, standard deviation = 1) within each participant as a method of feature scaling. The SVM was trained using 48 samples from 24 participants (i.e., the high and low encoding states) with the default parameters (kernel type = radial basis function, gamma = 1/the number of features; $c$ = 1), and tested using two samples from the left-out participant. The classification accuracy was averaged across the 25 folds of cross validation. The statistical significance of classification accuracy was evaluated using a permutation test, as proposed by *Golland and Fischl, 2003* (*Golland and Fischl, 2003*). In the permutation test, the class labels (i.e., the high or low encoding states) of the original data are reversed in randomly selected participants, and the same SVM classification was performed to obtain a null distribution of classification accuracy (10,000 permutations) The $P$ value was calculated as the proportion of classification accuracies that are equal to or greater than the accuracy obtained by the original data. For control analyses, we repeated the same classification procedure, except we used different sets of input variables.

## Acknowledgements

We thank Yusuke Noro for insightful discussion.

## Additional information

### Funding

| Funder | Grant reference number | Author |
|---|---|---|
| Japan Society for the Promotion of Science | 17H00891 | Ryuta Aoki<br>Koji Jimura<br>Kiyoshi Nakahara |
| Japan Society for the Promotion of Science | 17H06268 | Kiyoshi Nakahara |
| Japan Society for the Promotion of Science | 15K12777 | Kiyoshi Nakahara |

The funders had no role in study design, data collection and interpretation, or the decision to submit the work for publication.

### Author contributions

Ruedeerat Keerativittayayut, Ryuta Aoki, Conceptualization, Formal analysis, Investigation, Writing—original draft; Mitra Taghizadeh Sarabi, Formal analysis, Investigation; Koji Jimura, Formal analysis, Writing—review and editing; Kiyoshi Nakahara, Conceptualization, Supervision, Funding acquisition, Investigation, Writing—review and editing

### Author ORCIDs

Ruedeerat Keerativittayayut (iD) http://orcid.org/0000-0002-9660-4794
Ryuta Aoki (iD) http://orcid.org/0000-0003-0282-4348
Kiyoshi Nakahara (iD) http://orcid.org/0000-0001-6701-6216

### Ethics

Human subjects: All experimental procedures were approved by the Ethics Committee of Kochi University of Technology. Informed consent was obtained from all participants.

### Decision letter and Author response

Decision letter https://doi.org/10.7554/eLife.32696.sa1
Author response https://doi.org/10.7554/eLife.32696.sa2

## Additional files

### Supplementary files

- Supplementary file 1. Supplementary tables for additional analyses.

- Transparent reporting form

### Data availability

The data that support the findings of this study are openly available in Dryad Digital Repository (https://datadryad.org/).

The following dataset was generated:

| Author(s) | Year | Dataset title | Dataset URL | Database and Identifier |
|---|---|---|---|---|
| Keerativittayayut R, Ryuta A, Mitra TS, Koji J, Kiyoshi N | 2018 | Data from: Large-scale network integration in the human brain tracks temporal fluctuations in memory encoding performance | http://dx.doi.org/10.5061/dryad.35kg335 | Dryad, 10.5061/dryad.35kg335 |

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
