## [Decision Letter]

Thank you for submitting your article "Large-scale network integration in the human brain tracks temporal fluctuations in memory encoding performance" for consideration by *eLife*. Your article has been reviewed by three peer reviewers, and the evaluation has been overseen by a Reviewing Editor and Sabine Kastner as the Senior Editor. The following individual involved in review of your submission has agreed to reveal her identity: Jessica Cohen (Reviewer #3).

The reviewers have discussed the reviews with one another and the Reviewing Editor has drafted this decision to help you prepare a revised submission.

As you can read below, the reviewers had several positive comments about your manuscript. They think that your study is interesting, novel, and important. They also praised your control analyses and the MVPA analysis.

At the same time, they had several concerns that must be addressed in the revision. Instead of repeating the reviewers' comments, I will highlight few issues in which the reviewers converged and/or I find particularly important.

1) Both reviewer 1 and 2 commented about the use of the Power et al. atlas. Reviewer 1 thinks the use of the Power et al. nomenclature is excessive and asks you to justify better the choice of this atlas. Reviewer 2 suggests adding the hippocampus to this atlas, so you can confirm the hippocampus-occipital effect you found using the Kim's ROIs.

2) Reviewer 1 noted your excessive use of reverse inference, which I also found excessive in the Discussion section. Perhaps instead of speculating about the contributions of brain regions you did not specifically investigate, you could use the Discussion section to focus on several issues noted by the reviewers.

3) Reviewer 1 commented that the 30-sec time window is not appropriate given the fast dynamics of memory encoding. I agree with this point. I also agree with reviewer 2 that, because of the long time window, the difference in memory between high and low memory states is small (comment #5). You have to report the average number of HH and LL/Miss trials in the two states. It would seem there were about 6 trials and 3 fixations per window, which means that with a mean of HH trials around 50%, high and low memory states could differ in just one HH. I suggest that you redo the analyses with a shorter time-window and using a parametric analysis with tertiles or quartiles as suggested by reviewer 2.

4) Reviewer 1 commented that your point that DMN activity is associated with subsequent forgetting is not true for the hippocampus, whose activity is assumed to be part of the DMN but shows subsequent memory effects. You need to discuss this issue, which has been previously investigated (e.g., Huijbers et al., 2011). Personally, what I found surprising is that you link the DMN to subsequent forgetting in the Introduction, but you then find higher DMN inter-subnetwork integration for high than low memory states and do not mention this apparent inconsistency in the Discussion. I suggest you report standard event-related analyses to confirm you are getting the standard subsequent forgetting effects in the DMN, particularly ventral parietal and posterior cingulate in your study. If so, you would have an interesting dissociation between activity and connectivity which could enhance the study.

5) Reviewer 1 notes that your conclusion that high encoding state is characterized by long- rather than short-range functional connectivity is not supported by any statistical analysis. Independently, reviewer 3 suggested a method for addressing this issue: use Euclidean distance of edges. I think you should add this analysis and ideally report a 2 (range: long vs. short) x 2 (memory state: high vs. low) interaction.

6) In addition to these points, the reviewers had several other comments you should address, such as using other measures of sub-network interactions (see Wig, 2017), investigate the connectivity of SFE regions, do additional analysis to control for potential motion confounds, and consider potential.

[Editors' note: further revisions were requested prior to acceptance, as described below.]

Thank you for resubmitting your work entitled "Large-scale network integration in the human brain tracks temporal fluctuations in memory encoding performance" for further consideration at *eLife*. Your revised article has been favorably evaluated by Sabine Kastner (Senior Editor), a Reviewing Editor, and three reviewers.

The manuscript has been improved but there are some remaining issues that need to be addressed before acceptance, as outlined below:

Reviewer 1:

1) Add an ANOVA to the last paragraph of the subsection “FC patterns among memory-encoding-related regions”.

2) Qualify the use of "short" in "short time windows (~36 s)".

3) Emphasize the "breaking" of resting state functional connectivity pattern during high encoding (changes in inter-subnetwork connectivity as opposed to FC increases).

4) Do not lump the hippocampus in with the DMN (focus on specific regions).

5) Clarify that subnetwork integration contributes to encoding (not the "core" roles of networks).

*Reviewer 2:*

1) Report only the 32P+scrubbing results.

2) Do not use "moment-to-moment".

3) Discuss studies using shorter time-scales, such as Sadaghiani et al. (2015).

4) Look at DMN subnetworks (e.g., Fornito et al., 2012) or community-detection algorithms to determine if the DMN is separated into multiple subnetworks during memory.

5) In first paragraph of the subsection “Dynamic reconfiguration of a large-scale functional brain network”, but not the conclusion, says that there is a general increase in long-range and decrease in short-range connections distributed across many networks, rather than a uniform increase/decrease in FC across the entire network.

6) Correct "Cohen & D'Esposito, 2016".

*Reviewer 3:*

1) Provide additional data explaining why the supplemental analysis show modularity was greater for high than low encoding states when movement is properly controlled.

2) Re-run the models without redundant regressors.

3) Add missing statistical tests.

4) Why weren't the SVMs run with the other variables?

Full comments of the reviewers:

*Reviewer #1:*

1) The stats starting in the last paragraph of the subsection “FC patterns among memory-encoding-related regions” require an ANOVA. Specifically, when comparing hippocampal and occipital connectivity for SME vs. SFE, separate t-tests are not appropriate (Nieuwenhuis et al., 2011).

2) "By analyzing time-varying FC within short time windows (~36 s)".

As indicated in the past review, "short" here is really incorrect. Single neuron studies show SME effects on the order of hundreds of milliseconds (Viskontas et al., 2006). Such effects are also present at the network level during retrieval in the local field potential on the scale of hundreds of milliseconds (Watrous et al., 2013). Thus, the authors really need to clarify in more detail in the Discussion that "short" here really refers to a state rather than specific processing related to memory. I suggest adding one or two sentences in the Discussion and mentioning these papers briefly.

3) "These findings suggest a systematic reconfiguration of the large-scale functional brain network related to incidental encoding performance, rather than a uniform increase/decrease in FC across the entire network."

I still think the authors need to be clearer that their findings also suggest "breaking" of resting state related functional connectivity patterns during high vs. low encoding states. Thus, the findings would also seem to support the idea that inter-subnetwork connectivity is important to these memory states rather than just increases in FC within them. There is some mention of this in the Discussion, but I was surprised not to see this point emphasized in more detail.

4) "(e.g., the hippocampus and other regions in the DMN)"

The authors should be careful here. The Power network does not include the hippocampus and different authors seem to lump the hippocampus in with the DMN while others do not. I suggest removing the statements about the DMN here and focusing on those specific regions. The authors may also consider an influential paper by Rugg and Vilberg that makes a much better case for memory specific brain regions than the resting state literature (Rugg and Vilberg, 2013).

5) "In our case, the results from these two metrics convergently suggested the core roles of the subcortical, default-mode, and visual systems in incidental encoding of visual stimuli."

Again, I think the authors should be careful here. They also showed integration across subnetworks was important to successful encoding. This doesn't suggest the "core" roles of these subnetworks themselves but rather their integration with each other, at least in the context of the paradigm investigated here.

*Reviewer #2:*

1) I appreciate the inclusion of the section of the Results "Addressing possible concerns about motion confounds". While acknowledging potential confounds is important, given the improved methods to deal with motion beyond the 6 motion parameters plus WM/CSF as you report in your main analyses (8P), I cannot think of a reason to not simply report the 32P+scrubbing results. As you acknowledge, the results are quite similar across the two methods. However, the differences across the methods may be related to motion, especially given your findings that FD is related to behavior, and that it is related to global efficiency without the more rigorous nuisance regression. These pieces of evidence all point to likely spurious results when you do not aggressively account for motion and other artifacts. Thus, I think you should remove all results using the 8P method and only report results using the 32P+scrubbing method.

2) At the beginning of the Discussion, you write: "We demonstrated dynamic reconfiguration of a large-scale functional brain network associated with moment-to-moment fluctuations in encoding performance." I would rephrase that, since moment-to-moment implies volume-to-volume (i.e., on the order of your TR) and/or differences on the resolution of individual trials; by using 36s, non-overlapping windows, this is longer than "moment-to-moment".

3) Related to the timing, most of the literature you cite having done similar analyses (arousal, sustained performance across blocks of a perception task or working memory/Stroop task), include tasks or states that are thought to vary on longer timescales, while your Introduction is about subsequent memory, which is categorized on a trial-by-trial basis. While I find the results you show averaging across trials convincing and an important contribution to the literature, more of a discussion about the shorter-scale changes in your case would be relevant. As it is, you briefly mention the shorter time-scale in the results but do not bring it up again. As an example in the literature, the Sadaghiani et al. (2015) paper that you reference looks at a small number of volumes before each detected or missed stimulus and does so on a trial-by-trial basis. A method like that seems appropriate for truly looking at subsequent memory, and as such should be discussed.

4) With regard to your discussion of the involvement of the DMN, it is helpful that you now point out that some regions within the DMN are also related to memory. Why don't you look at sub-DMN networks, which has been done in other studies looking at memory network dynamics in the past (i.e., Fornito et al., 2012; and others)? Or, in the very least, suggest using community detection algorithms to determine whether the DMN is more accurately separated into multiple subnetworks during memory?

5) In the first paragraph of the Discussion subsection “Dynamic reconfiguration of a large-scale functional brain network”, I appreciate the increased specificity in your explanation. However, it selects only two examples (there were significant increases within more networks than just the DMN, for example) thus it misrepresents the results. Additionally, the specific examples in contrast to the summary of distance effects is confusing – the distance effects appear to be across the whole-brain and not related to individual networks, whereas you initiate the paragraph giving examples only of a small subset of individual networks. It seems as though a conclusion more in line with the results is that there is a general increase in long-range and decrease in short-range connections distributed across many networks, rather than a uniform increase/decrease in FC across the entire network.

6) Finally, as a small comment, this paper is cited incorrectly in the in-text citations: "Cohen and Esposito, 2016" is incorrect; it should be "Cohen and D'Esposito, 2016". I see it cited about 4-5 times, so it should be fixed each time.

*Reviewer #3:*

1) My first concern centers on the results following more appropriate strategies for dealing with movement, which is known to impact time-course correlations. I appreciate the effort that was put towards minimizing movement-related confounds. A large proportion of the analyses do not survive correction for multiple comparison corrections. More critically however, a supplemental analysis indicates that modularity is actually increased during high encoding states relative to low encoding states when movement is properly controlled. This is in large conflict with the remaining analyses, and is incompatible with the conclusions of the paper. As the paper is framed around segregation and integration, the fact that the closest measure to segregation exhibits an opposite pattern to that which is discussed is problematic. The authors discuss this point a little and offer a scenario where modularity and efficiency can exhibit opposing patterns (subsection “The effects of denoising methods”, last paragraph), but I'm not convinced that the disconnect between their measures has been reconciled. I believe additional work needs to be done to explore this discrepancy, possibly centering in on what parts of the network are driving the effect, as readers interested in the network side of things will question the basis for the conclusions.

2) For both the GLM and time-course analysis (the latter of which is used for all subsequent connectivity/graph comparisons), it appears all trial-types have been modeled explicitly (high-hit, low-hit, miss, fixation; subsection “Trial-related activation analysis”). As a result, I think the models contain redundant regressors which can impact estimation of the regression coefficients and residuals. This should be corrected.

3) A number of necessary statistical tests are missing to allow comparisons across states/measures. Specifically, for examining SME/SFE by state, the analysis of within subnetwork connectivity should reveal an interaction, as that is what is being implied and interpreted. It is currently presented as a series of pairwise comparisons (subsections “FC patterns among memory-encoding-related regions” and “FC patterns across large-scale brain networks”). Likewise, for local/global efficiency/PC vs. high/low encoding (subsection “Graph analysis on large-scale brain network”), a comparable ANOVA model is required to confirm the existence of interactions.

4) Why weren't SVMs run with the other variables? Is there a reason why the authors only report the results of PC, subnetwork local-e, and whole-matrix FC patterns (subsection “Multivariate pattern classification using graph metrics as features”)?

---

## [Author Response]

As you can read below, the reviewers had several positive comments about your manuscript. They think that your study is interesting, novel, and important. They also praised your control analyses and the MVPA analysis.At the same time, they had several concerns that must be addressed in the revision. Instead of repeating the reviewers' comments, I will highlight few issues in which the reviewers converged and/or I find particularly important.1) Both reviewer 1 and 2 commented about the use of the Power et al. atlas. Reviewer 1 thinks the use of the Power et al. nomenclature is excessive and asks you to justify better the choice of this atlas. Reviewer 2 suggests adding the hippocampus to this atlas, so you can confirm the hippocampus-occipital effect you found using the Kim's ROIs.

We thank the editor for summarizing these important comments provided by the reviewers. In accord with these comments, we intensively revised our manuscript. First, we avoided the excessive use of the Power et al. nomenclature in the revised manuscript (e.g., in Discussion). Second, we clarified why we chose the Power atlas, explaining that the atlas has been used in previous task-fMRI studies. Third, to avoid the over-reliance on the Power atlas, we cross-validated our results using an independent atlas (Gordon et al., 2016). Furthermore, in accordance with the comment by reviewer #2, we added the bilateral hippocampus ROIs to the Power atlas and confirmed the increased functional connectivity between the hippocampus and occipital cortex associated with memory encoding performance (Materials and methods, subsection “Regions of interest”; Results, subsection “Robustness check”, last paragraph and subsection “FC patterns across large-scale brain networks”, last paragraph).

2) Reviewer 1 noted your excessive use of reverse inference, which I also found excessive in the Discussion section. Perhaps instead of speculating about the contributions of brain regions you did not specifically investigate, you could use the Discussion section to focus on several issues noted by the reviewers.

In response to this comment, we removed the speculations based on reverse inferences (e.g., relating the SUB to intrinsic motivation and the CON to sustained attention). Instead, we added new paragraphs to Discussion to address the issues noted by the reviewers (e.g., Discussion, subsection “Network integration and segregation associated with incidental encoding performance”, last paragraph; subsection “Methodological considerations for graph analysis”, first and second paragraphs).

3) Reviewer 1 commented that the 30-sec time window is not appropriate given the fast dynamics of memory encoding. I agree with this point. I also agree with reviewer 2 that, because of the long time window, the difference in memory between high and low memory states is small (comment #5). You have to report the average number of HH and LL/Miss trials in the two states. It would seem there were about 6 trials and 3 fixations per window, which means that with a mean of HH trials around 50%, high and low memory states could differ in just one HH. I suggest that you redo the analyses with a shorter time-window and using a parametric analysis with tertiles or quartiles as suggested by reviewer 2.

We agree that the previous manuscript was not clear enough on these points. In the revised manuscript, we have made it clearer why we had an a priori interest in dynamic functional connectivity in the timescale of 30–40 s. In addition, we reported the number of HH trials per window in the high and low encoding states (high: 5.18 ± 1.25 trials, low: 2.59 ± 1.17 trials) in Results. Furthermore, we performed additional analyses with shorter time windows (e.g., 7.2 s) and using tertiles and quartiles, which confirmed that our results are robust against changes in these settings (Results, subsection “Classification of time windows based on encoding performance”, first paragraph; subsection “Robustness check”, first paragraph).

4) Reviewer 1 commented that your point that DMN activity is associated with subsequent forgetting is not true for the hippocampus, whose activity is assumed to be part of the DMN but shows subsequent memory effects. You need to discuss this issue, which has been previously investigated (e.g., Huijbers et al., 2011). Personally, what I found surprising is that you link the DMN to subsequent forgetting in the Introduction, but you then find higher DMN inter-subnetwork integration for high than low memory states and do not mention this apparent inconsistency in the Discussion. I suggest you report standard event-related analyses to confirm you are getting the standard subsequent forgetting effects in the DMN, particularly ventral parietal and posterior cingulate in your study. If so, you would have an interesting dissociation between activity and connectivity which could enhance the study.

In accordance with the comment by reviewer #1, we revised the manuscript and made a clear distinction between the SFE and the DMN (e.g., in Introduction and Discussion). That is, although some of the SFE regions (e.g., the posterior cingulate cortex and temporoparietal junction) are included in the DMN (Kim, 2011), the SFE regions and the DMN are not identical.

Moreover, following the suggestion by the editor, we performed event-related analyses and added new results to Results and panels to Figure 3. The results confirmed the subsequent forgetting effects in the SFE regions (including the posterior cingulate cortex and temporoparietal junction). Based on this observation, we added a discussion about the potential dissociations between trial-related activation and functional connectivity (Results, subsection “FC patterns among memory-encoding-related regions”, first paragraph; Discussion, subsection “Network integration and segregation associated with incidental encoding performance”, last paragraph).

5) Reviewer 1 notes that your conclusion that high encoding state is characterized by long- rather than short-range functional connectivity is not supported by any statistical analysis. Independently, reviewer 3 suggested a method for addressing this issue: use Euclidean distance of edges. I think you should add this analysis and ideally report a 2 (range: long vs. short) x 2 (memory state: high vs. low) interaction.

In response to the comments made by reviewers #1 and #3, we performed a new analysis using Euclidean distance of edges. The result revealed that the mean Euclidean distance of edges showing functional connectivity increases (high vs. low encoding state) was significantly longer than that of edges showing functional connectivity decreases. In other words, we confirmed “a difference (i.e., Euclidean distance) in differences (i.e., increases [high > low] vs. decreases [high < low] in functional connectivity)” as a proxy for the 2-by-2 interaction suggested by the editor (Results, subsection “FC patterns across large-scale brain networks”, first paragraph).

6) In addition to these points, the reviewers had several other comments you should address, such as using other measures of sub-network interactions (see Wig, 2017), investigate the connectivity of SFE regions, do additional analysis to control for potential motion confounds, and consider potential.

We carefully addressed these issues by performing an extensive set of additional analyses, as you can see below in our responses to each reviewer’s comments. We believe that the quality of our manuscript has been much improved by incorporating these valuable comments of the editor and reviewers.

[Editors' note: further revisions were requested prior to acceptance, as described below.]

Reviewer #1:1) The stats starting in the last paragraph of the subsection “FC patterns among memory-encoding-related regions” require an ANOVA. Specifically, when comparing hippocampal and occipital connectivity for SME vs. SFE, separate t-tests are not appropriate (Nieuwenhuis et al., 2011).

We thank the reviewer for this comment. In the revised manuscript, we included an additional ANOVA to test the interaction between state (high vs. low encoding states) and subnetwork (SME vs. SFE subnetworks) in our analysis of the within-subnetwork connectivity (subsection “FC patterns among memory encoding-related regions”, last paragraph).

We would like to note that, in the analysis comparing hippocampal and occipital connectivity, we tested FC difference (high vs. low encoding state) in the hippocampal-occipital node pair. As such, this test involved only one factor, and a two-way ANOVA was not applicable.

2) "By analyzing time-varying FC within short time windows (~36 s)".As indicated in the past review, "short" here is really incorrect. Single neuron studies show SME effects on the order of hundreds of milliseconds (Viskontas, Knowlton et al., 2006). Such effects are also present at the network level during retrieval in the local field potential on the scale of hundreds of milliseconds (Watrous, Tandon et al., 2013). Thus, the authors really need to clarify in more detail in the Discussion that "short" here really refers to a state rather than specific processing related to memory. I suggest adding one or two sentences in the Discussion and mentioning these papers briefly.

We agree with the reviewer that the timescale of our interest (i.e., 30–40 s) was not “short,” given the rapid nature of the neural processes involved in single memory-encoding events. In the revised manuscript, we avoided using the term “short” to describe our time windows. For instance, the description such as “short time windows” was replaced with “36-s time windows”.

Furthermore, we added several new sentences to clarify that our finding pertains to the link between particular states of large-scale networks and average memory performance during given time periods, rather than neural processes underlying single memory-encoding events (Discussion, subsection “Network integration and segregation associated with incidental encoding performance”, last paragraph). We also cited the papers suggested by the reviewer, and discussed that large-scale neural synchronizations in timescales of hundreds of milliseconds would also play key roles in both successful memory encoding and retrieval.

3) "These findings suggest a systematic reconfiguration of the large-scale functional brain network related to incidental encoding performance, rather than a uniform increase/decrease in FC across the entire network."I still think the authors need to be clearer that their findings also suggest "breaking" of resting state related functional connectivity patterns during high vs. low encoding states. Thus, the findings would also seem to support the idea that inter-subnetwork connectivity is important to these memory states rather than just increases in FC within them. There is some mention of this in the Discussion, but I was surprised not to see this point emphasized in more detail.

We thank the reviewer for this constructive suggestion. Although we did not acquire resting-state fMRI data and therefore could not directly compare the dynamic FC patterns during the high encoding state with those during task-free resting states, our results suggest that “breaking” resting-state FC patterns is important for successful memory encoding of externally presented visual stimuli. We have included several new sentences to emphasize this point (Discussion, subsection “Dynamic reconfiguration of a large-scale functional brain network”, first paragraph).

4) "(e.g., the hippocampus and other regions in the DMN)"The authors should be careful here. The Power network does not include the hippocampus and different authors seem to lump the hippocampus in with the DMN while others do not. I suggest removing the statements about the DMN here and focusing on those specific regions. The authors may also consider an influential paper by Rugg and Vilberg that makes a much better case for memory specific brain regions than the resting state literature (Rugg and Vilberg, 2013).

In accord with the reviewer’s comment, we removed the term “DMN” from this sentence and replaced it with specific regions (i.e., the posterior cingulate, ventral parietal, and medial prefrontal cortices) reported in the paper we cited (Discussion, subsection “Dynamic reconfiguration of a large-scale functional brain network”, last paragraph). These regions, together with the MTL, are also highlighted in Rugg and Villberg (2013) as a part of the “general recollection network.” To acknowledge their view, we cited Rugg and Villberg (2013) in the revised manuscript and have clarified that this network plays a key role in successful memory retrieval (see the aforementioned paragraph).

5) "In our case, the results from these two metrics convergently suggested the core roles of the subcortical, default-mode, and visual systems in incidental encoding of visual stimuli."Again, I think the authors should be careful here. They also showed integration across subnetworks was important to successful encoding. This doesn't suggest the "core" roles of these subnetworks themselves but rather their integration with each other, at least in the context of the paradigm investigated here.

We apologize that the description in the previous version of the manuscript was misleading. In the revised manuscript, we removed the term “core” and modified the text to “the results of these two metrics convergently suggested that integration of the SUB with other subnetworks is associated with successful memory encoding”.

Reviewer #2:1) I appreciate the inclusion of the section of the Results "Addressing possible concerns about motion confounds". While acknowledging potential confounds is important, given the improved methods to deal with motion beyond the 6 motion parameters plus WM/CSF as you report in your main analyses (8P), I cannot think of a reason to not simply report the 32P+scrubbing results. As you acknowledge, the results are quite similar across the two methods. However, the differences across the methods may be related to motion, especially given your findings that FD is related to behavior, and that it is related to global efficiency without the more rigorous nuisance regression. These pieces of evidence all point to likely spurious results when you do not aggressively account for motion and other artifacts. Thus, I think you should remove all results using the 8P method and only report results using the 32P+scrubbing method.

Following this suggestion, we revised the manuscript to report only the results using the 32P+scrubbing method. We believe that this modification has made our manuscript more readable.

Because of this change, some of the results became statistically non-significant after FDR correction (e.g., FC patterns in the SME/SFE regions). We have retained all results regardless of statistical significance, and added discussion about the possible reasons for the lack of statistical significance (subsection “Time-varying FC among memory-related brain regions”, last paragraph).

2) At the beginning of the Discussion, you write: "We demonstrated dynamic reconfiguration of a large-scale functional brain network associated with moment-to-moment fluctuations in encoding performance." I would rephrase that, since moment-to-moment implies volume-to-volume (i.e., on the order of your TR) and/or differences on the resolution of individual trials; by using 36s, non-overlapping windows, this is longer than "moment-to-moment".

We agree that the term “moment-to-moment” may imply a shorter timescale than we actually analyzed. We removed all uses of the term “moment-to-moment” from the revised manuscript and replaced them with more appropriate terms, such as “temporal fluctuations” (Discussion, first paragraph).

3) Related to the timing, most of the literature you cite having done similar analyses (arousal, sustained performance across blocks of a perception task or working memory/Stroop task), include tasks or states that are thought to vary on longer timescales, while your Introduction is about subsequent memory, which is categorized on a trial-by-trial basis. While I find the results you show averaging across trials convincing and an important contribution to the literature, more of a discussion about the shorter-scale changes in your case would be relevant. As it is, you briefly mention the shorter time-scale in the results but do not bring it up again. As an example in the literature, the Sadaghiani et al. (2015) paper that you reference looks at a small number of volumes before each detected or missed stimulus and does so on a trial-by-trial basis. A method like that seems appropriate for truly looking at subsequent memory, and as such should be discussed.

We agree with the reviewer’s point that trial-by-trial analysis of FC patterns in shorter timescales would be useful for examining single-item subsequent memory in detail. In the revised manuscript, we added a new discussion about our results using smaller time windows (at a minimum of 7.2 s [10 TRs]), and clarified that our analysis did not pertain to trial-by-trial dynamic FC patterns even with our shortest timescale. We also described trial-by-trial approach used in Sadaghiani et al. (2015) and suggested that their methods could be employed for memory encoding research in the future (Discussion, subsection “Network integration and segregation associated with incidental encoding performance”, last paragraph).

4) With regard to your discussion of the involvement of the DMN, it is helpful that you now point out that some regions within the DMN are also related to memory. Why don't you look at sub-DMN networks, which has been done in other studies looking at memory network dynamics in the past (i.e., Fornito et al., 2012; and others)? Or, in the very least, suggest using community detection algorithms to determine whether the DMN is more accurately separated into multiple subnetworks during memory?

Following the reviewer’s constructive suggestion, we performed group-level modular decomposition for the DMN subnetwork, using Fornito et al.’s (2012) method (subsection “Graph analysis”, tenth paragraph). We found that the DMN was further divided into five submodules. The submodule structure was identical between the high and low encoding states. When we examined “submodule-wise” graph metrics, we found a trend indicating higher submodule-wise local efficiency during the high vs. low encoding state in one of the submodules (consisting of nodes located in the medial PFC; *z*_(24)_ = 2.3274, *P* = 0.0199), but this was not significant after FDR correction. Although this result could imply that a specific DMN submodule is related to memory encoding, further research is needed to elucidate possible differentiation among the DMN submodules for memory encoding. We described these observations in the Results section (subsection “Graph analysis on large-scale brain network”, last paragraph) and the Discussion section (subsection “Network integration and segregation associated with incidental encoding performance”, second paragraph). We also added a new figure (Figure 7) to visualize the DMN submodule structure.

Because Fornito et al. (2012) used the Louvain algorithm to identify optimal module structure, we also used the same algorithm to analyze our data. To maintain consistency, we employed the Louvain algorithm throughout the manuscript to compute the optimal module structure and modularity (except for the modularity contribution analysis, which required pre-defined community labels from the Power atlas). Although the Louvain algorithm is stochastic, we confirmed that our results were stable over iterations (mean ± SD of *z* value [Wilcoxon signed-rank test] over 5,000 iterations = 2.6051 ± 0.0709 for the modularity difference between the high and low encoding states). In addition, modularity computed by the Louvain algorithm was highly correlated with that computed by Newman (deterministic) algorithm (window-to-window correlation: Pearson’s *r* = 0.9796 ± 0.0112; *z*_(24)_ = 4.3724, *P* = 1.2290 × 10^-5^, signed-rank test) (subsection “Graph analysis”, seventh paragraph).

5) In the first paragraph of the Discussion subsection “Dynamic reconfiguration of a large-scale functional brain network”, I appreciate the increased specificity in your explanation. However, it selects only two examples (there were significant increases within more networks than just the DMN, for example) thus it misrepresents the results. Additionally, the specific examples in contrast to the summary of distance effects is confusing – the distance effects appear to be across the whole-brain and not related to individual networks, whereas you initiate the paragraph giving examples only of a small subset of individual networks. It seems as though a conclusion more in line with the results is that there is a general increase in long-range and decrease in short-range connections distributed across many networks, rather than a uniform increase/decrease in FC across the entire network.

In accord with the reviewer’s comment, we removed the specific examples from this paragraph and placed them in a later section (subsection “Network integration and segregation associated with incidental encoding performance”, second paragraph). In the revised manuscript, we referred to the Supplementary file 1D and 1E to indicate the complete lists of connections showing FC changes (subsection “Dynamic reconfiguration of a large-scale functional brain network”, first paragraph). Following this modification, the revised paragraph is dedicated to discussing the findings illustrated in Figure 4C–E.

6) Finally, as a small comment, this paper is cited incorrectly in the in-text citations: "Cohen and Esposito, 2016" is incorrect; it should be "Cohen and D'Esposito, 2016". I see it cited about 4-5 times, so it should be fixed each time.

We apologize for this mistake. We have corrected all in-text citations.

Reviewer #3:1) My first concern centers on the results following more appropriate strategies for dealing with movement, which is known to impact time-course correlations. I appreciate the effort that was put towards minimizing movement-related confounds. A large proportion of the analyses do not survive correction for multiple comparison corrections.

We would like to emphasize that our main finding, the large-scale network integration during successful memory encoding, was significant not only in the main analysis but also across a range of robustness-check analyses (subsection “Robustness check”).

More critically however, a supplemental analysis indicates that modularity is actually increased during high encoding states relative to low encoding states when movement is properly controlled. This is in large conflict with the remaining analyses, and is incompatible with the conclusions of the paper. As the paper is framed around segregation and integration, the fact that the closest measure to segregation exhibits an opposite pattern to that which is discussed is problematic. The authors discuss this point a little and offer a scenario where modularity and efficiency can exhibit opposing patterns (subsection “The effects of denoising methods”, last paragraph), but I'm not convinced that the disconnect between their measures has been reconciled. I believe additional work needs to be done to explore this discrepancy, possibly centering in on what parts of the network are driving the effect, as readers interested in the network side of things will question the basis for the conclusions.

We thank the reviewer for this thoughtful comment. We agree with the reviewer that it appears counterintuitive for a dynamic network to exhibit both increased integration and modularity at the same time. However, previous theoretical studies have shown that networks can exist in such a paradoxical state (Tononi, Sporns, and Edelman 1994; Watts and Strogatz 1998; Pan and Sinha 2009; Meunier, Lambiotte, and Bullmore 2010). Moreover, empirical studies of brain networks have shown that integration and modularity can functionally coexist (Mohr et al., 2016; Bertolero et al., 2018).

In accord with the reviewer’s suggestion, we performed detailed analyses to clarify which parts of the network drove our finding regarding modularity.

First, we examined the relationship between modularity (entire network-level segregation) and global efficiency (entire network-level integration) by computing the window-to-window correlation within participants. We found that these two metrics were *not* anti-correlated with each other (Pearson’s *r* = 0.1155 ± 0.4408; *z*_(24)_ = 1.3319, *P* = 0.1829), suggesting that these two graph metrics captured somewhat independent aspects of the network architecture. Thus, our finding of higher modularity during the high encoding state was not necessarily in substantial conflict with the finding of global efficiency.

Second, we computed the modularity contribution of each subnetwork (Sadaghiani et al., 2015) (Materials and methods, subsection “Graph analysis”, eighth paragraph). We found a significant state-by-subnetwork interaction (*F*_(9, 216)_ = 3.4900, *P* = 0.0005, two-way ANOVA), revealing that the modularity-contribution differences between the states varied across the subnetworks. At the subnetwork level, we found trends indicating a higher modularity contribution during the high vs. low encoding state in three subnetworks, most notably in the SUB (SUB: *z*_(24)_ = 2.7311, *P* = 0.0063; DAN: *z*_(24)_ = 2.3274, *P* = 0.0199; DMN: *z*_(24)_ = 2.0584, *P* = 0.0396; Wilcoxon signed-rank test), although these results did not survive FDR correction, possibly due to the use of suboptimal (pre-defined) community labels in this analysis.

Third, to further clarify the trend of increased modularity contribution (high vs. low encoding state) in these subnetworks, we examined the proportion of edges within single subnetworks and those connecting different subnetworks. In the SUB, we found that the proportion of edges within the SUB was higher during the high encoding state, and that the proportion of edges between the SUB and the other subnetworks was higher during the high encoding state. In addition, the increase in the proportion of edges (high vs. low encoding states) was greater for the connections within the SUB than those between the SUB and the other subnetworks (*z*_(24)_ = 2.5965, *P* = 0.0094). A two-way ANOVA revealed an interaction (*F*_(1, 24)_ = 8.4671, *P* = 0.0077) between state (high vs. low) and connection type (within vs. across).

These results suggest that the SUB contributed to entire network-level segregation (i.e., modularity) by its increased within-subnetwork connections, and also contributed to inter-subnetwork integration by its increased connection with other subnetworks. Thus, coexistence of greater integration and greater modularity could be reconciled with the increased proportions of edges within the SUB itself and between the SUB and the other subnetworks. We have discussed these issues in the revised Results and Discussion sections (subsection “Graph analysis on large-scale brain network”, third and fourth paragraphs; subsection “Methodological considerations for graph analysis”, second paragraph).

However, although the modularity difference between the two states was statistically significant in our main analysis, we observed that it was not so robust against one of the two supplement analyses controlling for motion confounds compared with the global efficiency difference (e.g., in the analysis using a subsample that exhibited minimal motion confounds). In addition, another measure of segregation (i.e., local efficiency) did not show significant difference between the two states (subsection “Graph analysis on large-scale brain network”, first paragraph). Therefore, we put more emphasis on integration rather than segregation, and refrained from drawing conclusive results of modularity in the current study. Further methodological improvements are needed to effectively identify possible coexistence of integration and segregation in large-scale brain networks related to various cognitive functions including memory encoding.

2) For both the GLM and time-course analysis (the latter of which is used for all subsequent connectivity/graph comparisons), it appears all trial-types have been modeled explicitly (high-hit, low-hit, miss, fixation; subsection “Trial-related activation analysis”). As a result, I think the models contain redundant regressors which can impact estimation of the regression coefficients and residuals. This should be corrected.

In accord with this comment, we revised our GLM to include only three trial-related regressors (i.e., high-confidence hit, low-confidence hit, and miss) for all trial-related activation and time-course analyses (subsection “Trial-related activation analysis”). None of our findings was affected by this modification. It should be noted that we included all four trial-related regressors in our original analysis because we followed Wagner et al. (1998)’s method, which explicitly modeled fixation events in their GLM.

3) A number of necessary statistical tests are missing to allow comparisons across states/measures. Specifically, for examining SME/SFE by state, the analysis of within subnetwork connectivity should reveal an interaction, as that is what is being implied and interpreted. It is currently presented as a series of pairwise comparisons (subsections “FC patterns among memory-encoding-related regions” and “FC patterns across large-scale brain networks”). Likewise, for local/global efficiency/PC vs. high/low encoding (subsection “Graph analysis on large-scale brain network”), a comparable ANOVA model is required to confirm the existence of interactions.

We thank the reviewer for pointing this out. We performed a two-way repeated-measures ANOVA to test the interaction between state (high vs. low encoding states) and subnetwork (SME vs. SFE subnetworks), and added the result to the revised manuscript (subsection “FC patterns among memory encoding-related regions”, last paragraph). In addition, we added a series of ANOVAs testing the interaction between state (high vs. low encoding state) and metrics (integration vs. segregation) for both the entire-network analysis (global efficiency vs. local efficiency) and the subnetwork-wise analysis (subnetwork-wise PC vs. subnetwork local efficiency). These analyses confirmed the existence of interactions (i.e., only integration, but not segregation, differed between the states) (subsection “Graph analysis on large-scale brain network”, first paragraph).

4) Why weren't SVMs run with the other variables? Is there a reason why the authors only report the results of PC, subnetwork local-e, and whole-matrix FC patterns (subsection “Multivariate pattern classification using graph metrics as features”)?

We appreciate the reviewer’s helpful comment. Accordingly, in the revised manuscript, we added a new SVM result using the inter-subnetwork efficiency (*E_is_*) as the input (Results, subsection “Multivariate pattern classification using graph metrics as features”, first paragraph). *E_is_* was also able to classify the high and low encoding states (72%, *P* = 0.0029, permutation test).

It should be noted that the purpose of the multivariate SVM classification analysis was to examine whether a “concise” set of features abstracted from large-scale network architectures was sufficient to distinguish the high and low encoding states. Thus, in the previous version of the manuscript, we used the subnetwork-wise PC and subnetwork-wise local efficiency, which were our measures of interest defined at the subnetwork level. In addition, we used the whole-matrix FC patterns as the input of the SVM, by which we confirmed that the whole-matrix FC patterns (_224_C_2_ = 24,976 features) did suffer from the curse of dimensionality, unlike the subnetwork-wise PC and subnetwork-wise local efficiency (10 features). We did not use global efficiency, entire-network local efficiency, or modularity because these graph metrics are unidimensional variables.